



# Reaction between CH₃C(O)OOH (peracetic acid) and OH in the gas-phase: A combined experimental and theoretical study of the kinetics and mechanism

Matias Berasategui[1], Damien Amedro[1], Luc Vereecken[2], Jos Lelieveld[1] and John N. Crowley[1]

[1] Division of Atmospheric Chemistry, Max-Planck-Institute for Chemistry, 55128 Mainz, Germany
[2] Institute for Energy and Climate Research: IEK-8, Forschungszentrum Juelich, 52425 Juelich, Germany

*Correspondence to*: John N. Crowley (john.crowley@mpic.de)

**Abstract.** Peracetic acid (CH₃C(O)OOH) is one of the most abundant organic peroxides in the atmosphere, yet the kinetics of its reaction with OH, believed to be the major sink, have been studied only once experimentally. In this work we combine

a pulsed-laser photolysis kinetic study of the title reaction with theoretical calculations of the rate coefficient and mechanism. We demonstrate that the rate coefficient is orders of magnitude lower than previously determined, with an experimentally derived upper limit of $\leq 4 \times 10^{-14}$ cm³ molecule⁻¹ s⁻¹. The relatively low rate coefficient is in good agreement with the theoretical result of $3 \times 10^{-14}$ cm³ molecule⁻¹ s⁻¹ at 298 K, increasing to $\sim 6 \times 10^{-14}$ in the cold upper troposphere, but with associated uncertainty of a factor-two. The reaction proceeds mainly via abstraction of the peroxidic-hydrogen via a

relatively weakly bonded and short-lived pre-reaction complex, in which H-abstraction occurs only slowly due to a high barrier and low tunneling probabilities. Our results imply that the lifetime of CH₃C(O)OOH with respect to OH-initiated degradation in the atmosphere is of the order of one year (and not days as previously believed) and that its major sink in the free and upper troposphere is likely to be photolysis, with dry-deposition important in the boundary layer. Similar conclusions can be made for other, saturated peroxy-acids.

**1 Introduction**

The processes leading to the formation and loss of two classes of atmospheric trace-gases, organic acids and organic peroxides, have been the subject of numerous field, laboratory and model based investigations (Atkinson et al., 2006; Calvert et al., 2011; Gunz and Hoffmann, 1990; Jackson and Hewitt, 1999; Lee et al., 2000; Paulot et al., 2011; Reeves and Penkett, 2003). In comparison, ambient measurements of the acidic peroxide, peracetic acid (CH₃C(O)OOH) in the gas-phase are still

relatively scarce, though measurements in the boundary layer (Crowley et al., 2018; Fels and Junkermann, 1994; He et al., 2010; Liang et al., 2013; Phillips et al., 2013; Walker et al., 2006; Zhang et al., 2010) and from aircraft (Crounse et al., 2006; Wang et al., 2019) indicate that it is present throughout the troposphere where it is expected to be the 2ⁿᵈ-most abundant organic peroxide (after CH₃OOH). Like other organic peroxides, CH₃C(O)OOH can contribute to the formation and aging of secondary organic aerosol (Docherty et al., 2005), which enhances its removal through wet deposition.



Unlike its non-peroxidic analogue, $CH_3C(O)OH$ (acetic acid), the direct emission of $CH_3C(O)OOH$ by the biosphere has not been documented and its formation during biomass burning has not been reported (Andreae, 2019), although elevated $CH_3C(O)OOH$ mixing ratios have been observed in air-masses impacted by biomass-burning (Crowley et al., 2018; Phillips et al., 2013). Apart from leakage during industrial production and application as an indoor disinfectant (Henneken et al., 2006; Pacenti et al., 2010), the only significant source of atmospheric PAA in the atmosphere is the radical terminating

channel (R1a) reaction between the acetylperoxy and hydroperoxyl radicals.

$$CH_3C(O)O_2 + HO_2 \qquad \rightarrow \qquad CH_3C(O)OOH + O_2 \qquad\qquad\qquad (R1a)$$
$$\rightarrow \qquad CH_3C(O)OH + O_3 \qquad\qquad\qquad (R1b)$$
$$\rightarrow \qquad OH + CH_3O_2 + CO_2 \qquad\qquad\qquad (R1c)$$

The $CH_3C(O)O_2$ radical is formed in the degradation of acetaldehyde (mainly via reaction with OH), acetone and methyl-

glyoxal (both via photolysis), all of which are common secondary products of the degradation of biogenic and anthropogenic volatile organic compounds (VOC) including isoprene, monoterpenes, alkenes and alkanes. Globally, biogenic emissions account for > 60% of $CH_3C(O)O_2$ formation (Fischer et al., 2014).

The highest production rates of $CH_3C(O)OOH$ are thus expected in regions which are impacted by biogenic emissions in which $HO_2$ levels are high enough to compete with $NO_2$ (R2) and NO (R3) for reaction with $CH_3C(O)O_2$:

$$CH_3C(O)O_2 + NO_2 + M \qquad \rightarrow \qquad CH_3C(O)OONO_2 + M \qquad\qquad\qquad (R2)$$
$$CH_3C(O)O_2 + NO\ (+O_2) \qquad \rightarrow \qquad CH_3O_2 + CO_2 + NO_2 \qquad\qquad\qquad (R3)$$

As $CH_3C(O)OONO_2$ (peroxyacetyl nitric anhydride, PAN) is thermally unstable with respect to re-dissociation to reactants, high temperatures also favour $CH_3C(O)OOH$ formation.

Laboratory studies indicate that the overall rate coefficient ($k_1$) for reaction R1 (at 298 K) is $(2 \pm 1) \times 10^{-11}$ cm$^3$ molecule$^{-1}$ s$^{-1}$

and that $CH_3C(O)OOH$ is formed with a branching ratio ($k_{1a} / k_1$) of $0.37 \pm 0.09$ at this temperature (Atkinson et al., 2006; IUPAC, 2020). At lower temperatures, such as those found in the upper troposphere, the rate coefficient increases ($k_1$ (240 K) = $3.7 \times 10^{-11}$ cm$^3$ molecule$^{-1}$ s$^{-1}$) while the branching ratio to $CH_3C(O)OOH$ decreases: $k_{1a} / k_1$ (240 K) = 0.31.

As for other soluble organic acids and peroxides, deposition will be an important sink for $CH_3C(O)OOH$ in the boundary layer, where (in the absence of measurements) an exchange velocity comparable to that of $H_2O_2$ (which has a similar

solubility, (Crowley et al., 2018)) may be assumed, and which results in a local lifetime of several hours. Further, wet deposition, either by direct dissolution or through particle formation and subsequent scavenging by clouds and rain, additionally reduces its lifetime. In analogy to other peroxides, reaction with OH is believed to be the dominant gas-phase loss process for $CH_3C(O)OOH$.

$$CH_3C(O)OOH + OH \qquad \rightarrow \qquad H_2O + CH_3C(O)OO \qquad\qquad\qquad (R4a)$$
$$\rightarrow \qquad H_2O + CH_2C(O)OOH \qquad\qquad\qquad (R4b)$$

The master-chemical-mechanism (MCM v3.2: http://mcm.leeds.ac.uk/MC) presently uses an overall rate coefficient of $k_2 = 3.6 \times 10^{-12}$ cm$^3$ molecule$^{-1}$ s$^{-1}$, which is based on reactions of OH with other organic trace-gases containing the –OOH group





(Orlando and Tyndall, 2003) whereby abstraction of the peroxidic H-atom (R4a), is expected to dominate. A single study of the rate coefficient has been published to date (Wu et al., 2017), in which the authors, deriving a rate coefficient of $\approx 1 \times 10^{-11}$ cm$^3$ molecule$^{-1}$ s$^{-1}$ in a relative-rate study, confirm the dominance of the OH-sink. There are no experimental studies of the branching ratio for the reaction between OH and CH$_3$C(O)OOH, though a theoretical study indicates that $k_{4a}$ and $k_{4b}$ are comparable (Rypkema and Francisco, 2013). Absorption cross-sections of CH$_3$C(O)OOH in the actinic region ($\lambda > 320$ nm) are lower than those of e.g. CH$_3$OOH and H$_2$O$_2$ (Burkholder et al., 2015; IUPAC, 2020; Orlando and Tyndall, 2003) and noon-time, mid-latitude photolysis rate constants are $\approx$ 5-7 $\times 10^{-7}$ s$^{-1}$. Assuming noon-time OH radical densities of $2 \times 10^6$ molecule cm$^{-3}$ and $k_4 = 3.6 \times 10^{-12}$ cm$^3$ molecule$^{-1}$ s$^{-1}$ results in a first-order loss rate constant of $\approx 7 \times 10^{-6}$ s$^{-1}$, which implies that the photolysis of CH$_3$C(O)OOH is not a significant atmospheric sink compared to reaction with OH. The relative importance of the various atmospheric loss processes for CH$_3$C(O)OOH in the light of the present results are discussed in more detail in section 4.6.

Measurements of ambient CH$_3$C(O)OOH mixing ratios have been used to gain insight into peroxyradical chemistry in the boreal forest during summer (Crowley et al., 2018; Phillips et al., 2013) and also to constrain the atmospheric budget of acetaldehyde, which is an important precursor of CH$_3$C(O)OOH (Travis et al., 2020; Wang et al., 2019). Such studies require accurate estimates of the lifetime of atmospheric CH$_3$C(O)OOH and thus the rate coefficient for its reaction with OH. The conclusions reached by Wang et al. (2019) based on aircraft data taken remote from sources of CH$_3$CHO are valid if CH$_3$C(O)OOH is much shorter lived than PAN and has a comparable lifetime to CH$_3$CHO. Likewise, the concentrations of CH$_3$C(O)OOH modelled by Crowley et al. (2018) are partially dependent on the OH-rate coefficient for the title reaction. In addition, the degree to which the formation of CH$_3$C(O)OOH from the reaction between HO$_2$ and CH$_3$C(O)O$_2$ represents a permanent sink of peroxy radicals (and thus loss of oxidation capacity) depends on whether the reaction of CH$_3$C(O)OOH with OH (to reform organic radicals) can compete with deposition processes.

In the following, we describe the results of laboratory experiments and theoretical calculations that show that CH$_3$C(O)OOH is much less reactive towards OH than presently believed. We also report rate coefficients ($k_5$) for the reaction between OD + CH$_3$C(O)OOH (R5) which was required to examine the potential role of OH reformation, and rate coefficients ($k_6$, $k_7$) for the reaction between both OH and OD with CH$_3$C(O)OH (R6, R7) which were required to apply corrections for the presence of CH$_3$C(O)OH in the CH$_3$C(O)OOH samples:

| CH$_3$C(O)OOH + OD | $\rightarrow$ | products | (R5) |
| CH$_3$C(O)OH + OH | $\rightarrow$ | products | (R6) |
| CH$_3$C(O)OH + OD | $\rightarrow$ | products | (R7) |

Finally, we examine the site-specificity of the H-abstraction reaction (R4a versus R4b).



## 2 Experimental Methods

The laboratory kinetic studies of the title reactions used the method of pulsed laser photolytic (PLP) generation of OH
combined with real time detection based on pulsed laser induced fluorescence (LIF), whereby the concentrations of
$CH_3C(O)OOH$ and $CH_3C(O)OH$ were measured on-line using infra-red absorption spectroscopy. The set-up is illustrated in
Fig. 1.

### 2.1 PLP-LIF

The details of the experimental set-up have been described in detail previously (Wollenhaupt et al., 2000) and only a brief
description of the central features and modifications will be given here. The experiments were carried out in a double-
jacketed quartz reactor of volume ~500 cm$^3$, which was held at the desired temperature by circulating a 60:40 mixture of
ethylene glycol/water through the outer jacket. The pressure in the reactor was monitored with 10, 100 and 1000 Torr
capacitance manometers. For all experiments, the axial flow velocity in the reactor was kept roughly constant at ~10 cm s$^{-1}$
by adjusting the flow rate. As the ~8 mm wide laser beam was normal to the direction of flow, this ensured that a fresh gas
sample was available for photolysis at each laser pulse. Pulses of 248 nm laser light (~20 ns) for OH generation from $H_2O_2$,
$DONO_2$ and PAA were provided at 10 Hz by an excimer laser (Compex 205 F, Coherent) operated using KrF.

| | | | |
|---|---|---|---|
| $H_2O_2 + h\nu$ | → | 2 OH | (R8) |
| $DONO_2 + h\nu$ | → | $OD + NO_2$ | (R9) |
| $CH_3C(O)OOH + h\nu$ | → | $CH_3C(O)O + OH$ | (R10) |

The laser fluence (typically ~15 mJ cm$^{-2}$ per pulse) was measured using a calibrated joule meter located behind the exit
window of the reactor. Concentrations of the OH-precursors, $H_2O_2$, $DONO_2$ and $CH_3C(O)OOH$ were in the range ~2–20 ×
$10^{13}$, 4–8 × $10^{14}$ molecule cm$^{-3}$, and ~6–60 × $10^{14}$ molecule cm$^{-3}$, respectively (see Tables S1 and S2). The initial OH
concentrations were calculated using 248 nm cross-sections (units of cm$^2$ molecule$^{-1}$) from Vaghjiani and Ravishankara
(1989a) ($\sigma_{248nm}(H_2O_2) = 9.3 \times 10^{-20}$,) Burkholder et al. (1993) ($\sigma_{248nm}(HNO_3) = 2.0 \times 10^{-20}$) and Orlando and Tyndall (2003)
($\sigma_{248nm}(CH_3C(O)OOH) = 3.41 \times 10^{-20}$ and were ~2–20 × $10^{11}$ molecule cm$^{-3}$.

Radiation for excitation of the OH $A^2\Sigma(\upsilon' = 1) \leftarrow X^2\Pi(\upsilon'' = 0)$ transition (Q11(1)) at 281.99 nm and OD $A^2\Sigma(v' = 1) \leftarrow$
$X^2\Pi(v'' = 0)$ transition at 287.68 nm was generated by a tuneable (YAG-pumped) dye laser (Quantel Brilliant B and Lambda
Physik Scanmate). The fluorescence of OH and OD was detected using a photomultiplier tube screened by a 309 nm
interference filter and a BG 26 glass cut-off filter. The fluorescence signal of OH was accumulated using a boxcar integrator
triggered at different delay times prior to and after the 248 nm laser to build up a time-dependent concentration profile.

### 2.2 On-line optical absorption measurements

The experiments to determine the rate coefficient of the title reaction were performed under pseudo-first-order conditions
(i.e. [acid]$_0$ >> [OH]$_0$) and the overall uncertainty in the rate constants was determined largely by the accuracy with which



the concentration of the excess reagent was measured. The concentrations of CH$_3$C(O)OH and CH$_3$C(O)OOH were therefore

continuously measured upstream of the reactor by flowing the sample through a 45 cm long absorption cell made of glass, which was equipped with silicon windows for transmission of infra-red (IR) light and a port for pressure measurement (using the same pressure gauges mentioned in section 2.1). With this set-up, absorption features in the range 600–4000 cm$^{-1}$ were constantly monitored (2 cm$^{-1}$ resolution, 16 co-added interferograms with 128 scans for the background) using a Fourier-Transform Infrared (FTIR) spectrometer (Bruker Vector 22) with an external, liquid-N$_2$ cooled HgCdTe detector. A low

spectral resolution was chosen to reduce scan times (~20 s) and enable rapid changes in concentration to be followed. OPUS software was used to analyse and manipulate the IR spectra. Interferograms were phase-corrected (Mertz) and Boxcar apodized with a zero-filling factor of 4. The concentrations of CH$_3$C(O)OH and CH$_3$C(O)OOH were calculated using reference spectra obtained in this work (see Section 2.2).

A further, in-line optical absorption systems was located downstream of the reactor. An absorption cell operated at 184.95

nm ($l$ = 43.8 cm, low pressure Hg lamp screened by a 185 nm interference filter) served to detect H$_2$O The VUV-absorption optical system is "dual beam" so that drifts in light intensity were accounted for and low optical densities could be measured over extended periods.

## 2.2 Off-line IR spectrum measurements

Reference spectra for CH$_3$C(O)OOH and CH$_3$C(O)OH were obtained with the Bruker Vector 22 coupled to two further IR-

absorption cells. These were a 44.39 L cylindrical quartz chamber equipped with a White-type, multiple-reflection mirror system with an 86.3 m optical path and external (HgCdTe) detector at liquid-N$_2$ temperature (Berasategui et al., 2020; Bunkan et al., 2018) and a 570 mL glass cell with a 15 cm optical path, located in the internal optical path of the FTIR using an internal DTGS detector. The pressure in both absorption cells was monitored using 1000 or 100 Torr capacitance manometers.

## 145  2.3 Chemicals

N$_2$ (Westfalen 99.999 %) was used without further purification. H$_2$O$_2$ (AppliChem, 50 wt.%) was concentrated to > 90% wt.% by vacuum distillation. Anhydrous DONO$_2$ was prepared by partial vacuum distillation of deuterated nitric acid formed by the addition of D$_2$SO$_4$ to KNO$_3$. Liquid CH$_3$C(O)OH and CH$_3$C(O)OOH solution (32 wt.% in acetic acid) were used following purification by partial vacuum distillation.

## 150  3 Theoretical Methods

The potential energy surface of the title reaction was characterized first at the M06-2X/cc-pVDZ level of theory (Dunning, 1989; Zhao and Truhlar, 2008). An exhaustive search for all conformers of reactants, transition states and products was performed by systematic variations of the dihedral angles for the degrees of freedom for internal rotation. Likewise, we





attempted to find all conformers of the pre- and post-reaction complexes by positioning the two complexing compounds at a
wide variety of relative orientations, and optimizing the geometry from each of these starting positions. All geometries were
subsequently re-optimized at the M06-2X-D3/aug-cc-pVTZ level of theory (Dunning, 1989; Goerigk et al., 2017; Grimme et
al., 2011), improving the description of the long-distance interactions by including diffuse orbitals and dispersion
corrections. The energy of the transition state for the abstraction of the per-acidic H-atom proved to be more dependent on
the basis set than expected, changing by ~4.3 kcal mol$^{-1}$ as opposed to ~2 kcal mol$^{-1}$ for the methyl H-abstraction, so to
ensure full convergence with respect to the basis set, we re-optimized all structures again at the M06-2X-D3/aug-cc-pVQZ
level of theory, confirming basis set convergence at the DFT level within a few tenths of a kcal mol$^{-1}$, and no significant
change in the geometries. ZPE corrections are done at this level, with vibrational wavenumbers scaled by 0.971 (Alecu et al.,
2010; Bao et al., 2018). Finally, the relative energies were refined using CCSD(T) single point calculations (Purvis and
Bartlett, 1982), extrapolated to the complete basis set using the aug-Schwartz6(DTQ) method by Martin (1996), based on
calculations using the aug-cc-pVxZ (x=D,T,Q) basis sets. The $T_1$ diagnostics do not suggest multi-reference character in any
of the structures. The strong dependence on the basis set size is assumed to be the main reason for the difference between our
barrier heights and those predicted by Rypkema and Francisco (2013) who found submerged transition states, incompatible
with the experimental data.

The temperature-dependent rate coefficient of the reaction was calculated using multi-conformer canonical transition state
theory (Truhlar et al., 1996; Vereecken and Peeters, 2003), including all conformers of reactants and transition states
characterized at our highest level of theory. The spin-orbit splitting for OH of 27.95 cm$^{-1}$ was taken into account (Huber and
Herzberg, 1979). Tunneling was accounted for by asymmetric Eckart barrier corrections based on the reactant, TS and
product energy, and the TS imaginary frequency (Eckart, 1930; Johnston and Heicklen, 1962). The rate coefficient is
calculated in the high-pressure limit; specifically, the pre-reaction complex is assumed to be in thermal equilibrium with the
reactants. Given the slow product formation rate, the protruding reaction barriers, and the fast formation and decomposition
of the complex this assumption seems reasonable. The main impact of the pre-reaction complex on the kinetics is then to
allow additional tunneling through a wider energy range of the TS barrier for H-abstraction. This is discussed in more detail
below.

## 4 Results and Discussion

**4.1 Infrared absorption Cross-Sections**

Accurate IR-absorption cross-sections of $CH_3C(O)OH$, its dimer and $CH_3C(O)OOH$ are required to derive accurate
concentrations of reactants during kinetic experiments on OH + $CH_3C(O)OOH$ where both acids are unavoidably present
because the commercially available sample of $CH_3C(O)OOH$ is a ~32% solution in $CH_3C(O)OH$.



### 4.1.1 CH₃C(O)OH and CH₃C(O)OH-dimer

In order to obtain the cross-sections of the $CH_3C(O)OH$ monomer, the long-path cell was used in conjunction with low pressures of $CH_3C(O)OH$ to avoid the formation of the dimer. A known pressure of the $CH_3C(O)OH$ sample (in total typically 3-18 Torr) was first measured in a section of the vacuum line (volume 126.6 cm³) and then flushed into the long-path cell in a flow of $N_2$ until 700 Torr of total pressure was reached. The pressure of $CH_3C(O)OH$ in the long-path cell (volume 44390 cm³) was calculated from the sum of monomer pressure plus twice the dimer pressure in the vacuum-line

(both calculated from the total pressure using the 298 K equilibrium coefficient $K_{eq} = 2.5 \pm 0.3$ Torr⁻¹ (Crawford et al., 1999)) and the dilution factor which is related to the volumes of the mixing line and the long-path cell. The concentration of $CH_3C(O)OH$ in the long-path cell was $1 - 10 \times 10^{14}$ molecule cm⁻³, where the equilibrium dimer concentration can be considered negligible. The $CH_3C(O)OH$-dimer spectrum was measured in the small cell (path-length 15 cm) using a similar procedure but using higher pressures of $CH_3C(O)OH + CH_3C(O)OH$-dimer (up to 1.36 Torr) to favour dimer formation. At

the highest total pressure, the ratio of dimer to monomer (calculated using the equilibrium constant listed above) was 1.4.

The $CH_3C(O)OH$ spectrum (Fig. 2) reveal features at 991, 1185, 1279, 1385, 1790 and 3583 cm⁻¹, with only the band at 3583 cm⁻¹ free of overlap with any of the $CH_3C(O)OOH$ or $CH_3C(O)OH$-dimer bands. The spectra obtained for $CH_3C(O)OH$ and $CH_3C(O)OH$-dimer are in excellent agreement with those available in the literature: At 1117 cm⁻¹ we derive $\sigma(CH_3C(O)OH$

$= 6.0 \times 10^{-19}$ cm² molecule⁻¹ which can be compared to $\sigma(CH_3C(O)OH = 5.9 \times 10^{-19}$ cm² molecule⁻¹ reported by Crawford et al. (1999). Similarly, our value of $\sigma(CH_3C(O)OH$-dimer) $= 1.8 \times 10^{-18}$ cm² molecule⁻¹ at 1295 cm⁻¹, is identical to that reported by Crawford et al. (1999).

Beer-Lambert plots for the 3583 cm⁻¹ band of $CH_3C(O)OH$ and the 1734 cm⁻¹ band of $CH_3C(O)OH$-dimer were constructed by expanding different pressures of $CH_3C(O)OH$ into the long-path cell at room temperature and 700 Torr of total pressure.

The results, displayed in Fig. S1 indicate a strictly linear relationship between band intensity and concentration over the range of concentrations investigated.

### 4.1.2 CH₃C(O)OOH

The liquid sample (32% (wt.) $CH_3C(O)OOH$ in $CH_3C(O)OH$) is prepared commercially by the oxidation of $CH_3C(O)OH$ using $H_2O_2$ and is a 4 component, equilibrium mixture:

$$CH_3C(O)OH(l) + H_2O_2(l) \qquad \rightleftharpoons \qquad H_2O(l) + CH_3C(O)OOH(l) \qquad\qquad\qquad (R11)$$

In order to obtain a quantitative IR-spectrum of $CH_3C(O)OOH$, head-space mixtures were dosed into the mixing line, where the total pressure ($CH_3C(O)OOH + CH_3C(O)OH + CH_3C(O)OH$-dimer $+ H_2O + H_2O_2$) was recorded before the mixture was flushed into the long-path cell. At the low concentrations of $CH_3C(O)OH$ in the long-path cell, there is no evidence for $CH_3C(O)OH$-dimer. In the absence of any absorption features of $H_2O_2$, the IR-absorption due to $CH_3C(O)OH$ and $H_2O$ was

converted to a vacuum line pressure of $CH_3C(O)OH + CH_3C(O)OH$-dimer $+ H_2O$, and the residual pressure assigned to



CH$_3$C(O)OOH, enabling quantification of the CH$_3$C(O)OOH spectrum (Fig. 2). As noted by (Crawford et al., 1999), there was no evidence for dimerization of CH$_3$C(O)OOH. These experiments were repeated using the 45 cm path-length absorption cell, which has the disadvantage that significantly higher concentrations of CH$_3$C(O)OOH are needed to obtain good quality spectra and the CH$_3$C(O)OH-dimer is therefore also present. However, it provides the advantage of eliminating

uncertainty related to the optical path-length. Additionally, using this set-up we obtained an accurate IR absorption spectra of H$_2$O relative to its VUV-absorption 185 nm using a cross-section of $7.1 \times 10^{-20}$ cm$^2$ molecule$^{-1}$ (Cantrell et al., 1997).

Using both set-ups, we derived a CH$_3$C(O)OOH cross-section at 1251 cm$^{-1}$ of $3.8 \times 10^{-19}$ cm$^2$ molecule$^{-1}$, with an uncertainty of 15%. This analysis neglects the contribution of H$_2$O$_2$ to the total head-space pressure. In offline experiments whereby the headspace was sampled into an enzyme/fluorescence based instrument for determination of H$_2$O$_2$ and organic peroxides

(Fischer et al., 2015) we found that H$_2$O$_2$ was present at about 1% of the CH$_3$C(O)OOH concentration, consistent with the very low vapour pressure of H$_2$O$_2$ compared to CH$_3$C(O)OOH.

Our CH$_3$C(O)OOH absorption cross-sections are much larger (factor 2.1) than those reported by Crawford et al. (1999) whose spectrum has $\sigma$(CH$_3$C(O)OOH) = $1.81 \times 10^{-19}$ cm$^2$ molecule$^{-1}$ at 1251.5 cm$^{-1}$. The only other published IR cross-sections of CH$_3$C(O)OOH of which we are aware were reported by Orlando et al. (2000a) in which a value of $5.3 \times 10^{-19}$ cm$^2$

molecule$^{-1}$ at 1251 cm$^{-1}$ is reported (~40% larger than our value), which was subsequently confirmed by the same group by comparison with HPLC measurements (Hasson et al., 2004). Note that both Orlando et al. (2000a) and Hasson et al. (2004) mistakenly listed this cross-section as being at 1295 cm$^{-1}$ instead of 1251 cm$^{-1}$, which was confirmed in private communication with the authors, who kindly provided their spectrum. Our spectrum, that of Orlando et al. (2000a) and one obtained by digitising Fig. 2 of Crawford et al. (1999) are displayed in Fig S3.

We do not have an explanation for the divergent values of the IR absorption spectrum of CH$_3$C(O)OOH, but note that this will, in part, be related to working with a multi-component mixture that requires accurate determination of the contributions of H$_2$O, CH$_3$C(O)OH and CH$_3$C(O)OH-dimer. As our experimental result could be reproduced in a series of experiments in two different experimental set-ups we use our own cross-sections to calculate CH$_3$C(O)OOH concentrations and consider the use of the larger value when estimating potential uncertainty in our rate coefficients. A Beer-Lambert plot for the 3306 cm$^{-1}$

band of CH$_3$C(O)OH (which we used to determine its concentration in kinetic experiments) is displayed in Fig. S1. As for CH$_3$C(O)OH and CH$_3$C(O)OH-dimer, the integrated band intensity was strictly proportional to concentration.

### 4.2 OH + CH$_3$C(O)OH: Determination of $k_6$ and $k_7$ at 298 K

We show later (section 4.3) that the reaction of OH with CH$_3$C(O)OH (R6) contributes to OH losses in the experiments designed to derive the rate coefficient for the title reaction and accurate rate coefficients under our experimental conditions

are necessary to account for this. We therefore carried out a set of experiments to measure the rate coefficient ($k_6$) for the reaction between OH and OD with CH$_3$C(O)OH.





In these experiments, OH was generated by the photolysis of $H_2O_2$ (0.3 - 1.8 × $10^{14}$ molecule cm$^{-3}$) and OD was generated by the photolysis of $DONO_2$ (5 × $10^{14}$ molecules cm$^{-3}$), both at 248 nm. Initial hydroxyl radical concentrations were $[OH]_0 \approx 1$ - 6 × $10^{11}$ and $[OD]_0 \approx 2 \times 10^{11}$ molecules cm$^{-3}$. The PLP-LIF studies were thus carried out under pseudo first-order conditions with $[CH_3C(O)OH] \gg [OH]$, so that the OH profiles are described by:

$$[OH]_t = [OH]_0 e^{-k't} \qquad (1)$$

where $[OH]_t$ is the concentration (molecule cm$^{-3}$) at time $t$ after the laser pulse and $k'$ is the pseudo-first-order rate coefficient and is defined as:

$$k' = k_6 [CH_3C(O)OH] + k_d \qquad (2)$$

where $k_6$ is the bimolecular rate coefficient (cm$^3$ molecule$^{-1}$ s$^{-1}$) for the reaction between OH and $CH_3C(O)OH$. $k_d$ (s$^{-1}$) accounts for OH loss due to transport out of the reaction zone and reaction with $H_2O_2$ or $DONO_2$.

Figure 3 displays representative data obtained at 295 K in $N_2$ bath gas for reaction of OH and OD with $CH_3C(O)OH$. The OH (OD) decays are strictly exponential and the plots of $k'$ versus $[CH_3C(O)OH]$ (Fig. 4) are straight lines as expected from Eq. 2. This analysis assumes that reaction of OH or OD with $CH_3C(O)OH$-dimer does not contribute significantly to its loss. In our experiments, the $CH_3C(O)OH$-dimer / $CH_3C(O)OH$ ratio in the reactor varied between 0.04 and 0.32. The strict linearity observed when plotting the first-order loss constant of OH or OD versus $[CH_3C(O)OH]$ is consistent with an insignificant contribution of $CH_3C(O)OH$–dimer to loss of OH, as also concluded by Singleton et al. (1989).

The values of $k_6$ and $k_7$ derived from these datasets typically have a statistical uncertainty ($2\sigma$) of less than 5%, so that the overall uncertainty is dominated by potential systematic error in the determination of $CH_3C(O)OH$ concentration, i.e. in the in-situ measurement of IR-absorption by $CH_3C(O)OH$ and its absorption cross-sections. During measurement of OH / OD decay, the $CH_3C(O)OH$ concentration was measured between 5 to 8 times using the 45 cm IR cell located upstream of the reactor and a small correction (~3%) for pressure differences between the IR-absorption cell and the reactor applied. Typically, the values of $[CH_3C(O)OH]$ varied by < 3% during the time required to measure the OH-decay, and therefore did not contribute significantly to overall uncertainty.

Experimental details (e.g. identity and concentration of OH precursor and pressure) as well as the values of the rate constants $k_6$ and $k_7$ at 298 K are presented in Table S1. We obtained values of $k_6 = (6.95 \pm 0.08) \times 10^{-13}$ cm$^3$ molecule$^{-1}$ s$^{-1}$ at 100 Torr total pressure and $k_6 = (7.04 \pm 0.28) \times 10^{-13}$ cm$^3$ molecule$^{-1}$ s$^{-1}$ at 250 Torr. The result for OH is thus in excellent agreement (2%) with the 298 K value of $6.9 \times 10^{-13}$ cm$^3$ molecule$^{-1}$ s$^{-1}$ presently recommended by the IUPAC panel (IUPAC, 2019). The IUPAC panel recommendation carries an uncertainty of $\pm$ 25% ($\Delta \log k = 0.1$) at 298 K. The very close agreement with our data may indicate that an uncertainty of $\pm$ 12% ($\Delta \log k = 0.05$) is more reasonable, and in the calculations below we use the IUPAC recommended expression $k_4 = 8.40 \times 10^{-20} T^2 \exp(1356/T)$ cm$^3$ molecule$^{-1}$ s$^{-1}$.

For the reaction between OD and $CH_3C(O)OH$, we obtain $k_7 = 7.3 \pm 0.3 \times 10^{-13}$ cm$^3$ molecule$^{-1}$ s$^{-1}$ at 298 K and a pressure of 125 Torr $N_2$, i.e. within 5% of the values obtained for OH. We are unaware of other measurements of this parameter with which to compare our value.





### 4.3 OH + CH₃C(O)OOH: Determination of $k_4$ and $k_5$ (298-353 K)

The experiments to measure $k_4$ were performed as described in section 4.2 for $CH_3C(O)OH$ with the difference that it was not necessary to add $H_2O_2$ as OH precursor, as the photolysis of $CH_3C(O)OOH$ itself provided sufficient OH. Taking a 248 nm laser fluence of ∼ 12 mJ cm$^{-2}$ per pulse, a 248 nm cross-section of $\sigma(CH_3C(O)OOH) = 3.4 \times 10^{-20}$ cm$^2$ molecule$^{-1}$ (Orlando and Tyndall, 2003) and assuming unity quantum yield we calculate that $[OH]_0$ varied between ∼ 3 - 20 × 10$^{11}$ molecule cm$^{-3}$ when the $CH_3C(O)OOH$ concentration was varied within the range 6.17 - 38.5 × 10$^{14}$ molecule cm$^{-3}$.

IR-absorption due to $CH_3C(O)OOH$, $CH_3C(O)OH$ and $CH_3C(O)OH$-dimer was monitored online in the 45 cm absorption cell (at 298 K). The concentrations of $CH_3C(O)OOH$, $CH_3C(O)OH$ and $CH_3C(O)OH$-dimer were quantified by scaling reference spectrum (sections 4.1.1 and 4.1.2) of each component to the measured composite spectrum as illustrated in Fig. S3. Typically, the concentrations of $CH_3C(O)OH$ vary between 3 × 10$^{14}$ and 2 × 10$^{15}$ molecule cm$^{-3}$ and those for $CH_3C(O)OOH$ between 6 × 10$^{14}$ and 6 × 10$^{15}$ molecule cm$^{-3}$. When the reactor is operated at high temperatures some of the $CH_3C(O)OH$-dimer present in the IR-absorption cell is converted to $CH_3C(O)OH$ in the reactor and correction was made to account for this.

The pseudo first-order conditions for both $[CH_3C(O)OOH] \gg [OH]$ and $[CH_3C(O)OH] \gg [OH]$ are thus guaranteed and the decay of OH is described by:

$$[OH]_t = [OH]_0\, e^{-(k_4' + k_6' + k_d)\, t} \tag{3}$$

Where $k_6'$ and $k_4'$ are the pseudo-first order rate constants for loss of OH via reaction (R6) and (R4), respectively.

Typical OH decays (at 298 and 353 K) in the presence of $CH_3C(O)OH$ and $CH_3C(O)OOH$ are displayed in Fig. 5a. As expected, the initial OH concentration varies with $[CH_3C(O)OOH]$ and OH decays are strictly exponential. The contribution of $CH_3C(O)OH$ to the decay of OH can be calculated as $k_6[CH_3C(O)OH]$. For this purpose, we use the IUPAC recommended parameterisation of $k_6$, the accuracy of which we have validated at 298 K as described above.

When $k_6[CH_3C(O)OH]$ is subtracted from the total first-order loss rate constant, we obtain $k_4[CH_3C(O)OOH] + k_d$. The rate coefficient $k_4$ can thus be derived from the slope of plots of $k_4[CH_3C(O)OOH] + k_d$ versus $[CH_3C(O)OOH]$ as illustrated in Fig. 6, which contains the data obtained at all temperatures. A least-squares fit to the entire dataset yields $k_4 = (3.25 \pm 0.46) \times 10^{-14}$ cm$^3$ molecule$^{-1}$ s$^{-1}$, independent of temperature. The complete dataset, listing the experimental conditions and the contribution of $CH_3C(O)OH$ to the total OH decay constant is found in Table S2. The uncertainty associated with the rate constant $k_4$ (listed in Table S2 and plotted in Fig. 6) considers the statistical error in deriving $k_6'$ and $k_4'$, as well as the uncertainty in the concentration of $CH_3C(O)OH$ (10-15%) (which is larger at high $[CH_3C(O)OH]$ owing to uncertainty in the dimer-monomer ratio, i.e. in $K_{eq}$) and in the rate coefficient $k_6$ (12%, see above). It does not consider systematic error $[CH_3C(O)OOH]$, which is discussed below in deriving the final value for $k_4$ and its total uncertainty.



### 4.4 Potential for systematic error in determining $k_4$ and $k_5$ (298-353 K)

The values we obtain for $k_4$ are clearly much lower (by a factor > 200) than the one previous relative-rate determination of (Wu et al., 2017) who report a room temperature rate coefficient of ~ $1 \times 10^{-11}$ cm$^3$ molecule$^{-1}$ s$^{-1}$. Below, we examine potential sources of systematic bias in our experiments and analysis.

#### 4.4.1 Uncertainty in the IR cross-sections of PAA

The accuracy of rate constants measured using the PLP-PLIF method under pseudo-first-order conditions depends predominantly on the accuracy of the measurement of the excess reagent, in this case $CH_3C(O)OOH$. Any systematic error in the IR cross-sections used to calculate [PAA] propagate directly into a systematic error in $k_4$. Although our measurements of the IR cross-sections of $CH_3C(O)OOH$ were in good agreement, irrespective of the absorption cell used, we noted divergence between our value and those previously published (see section 4.1.2). For this reason, we expand the uncertainty on our cross-sections to ± 25% so that the results agree (within combined experimental uncertainty) with those reported by (Orlando et al., 2000b). This implies an additional uncertainty of 25% for $k_4$

#### 4.4.2 Reformation of OH

A possible cause of a low rate coefficient measured in our direct study is the reformation of OH via decomposition of a reaction product, as has been observed (Vaghjiani and Ravishankara, 1989b) in the reaction of OH with another organic peroxide, $CH_3OOH$ (R12a, R13):

$$OH + CH_3OOH \qquad \rightarrow \qquad H_2O + CH_2OOH \qquad \qquad (R12a)$$
$$\rightarrow \qquad H_2O + CH_3O_2 \qquad \qquad (R12b)$$
$$CH_2OOH \qquad \rightarrow \qquad OH + CH_2O \qquad \qquad (R13)$$

In analogy, if the decomposition to OH of any reaction product of $CH_3C(O)OOH$ + OH were sufficiently rapid, our experiment would underestimate the rate coefficient. In order to rule this out, we conducted experiments in which OH was replaced with OD. In this case, the reformation of OH via dissociation of the O-OH bond would not impact on the kinetic measurement.

The results of experiments in which the 248 nm photolysis of $DONO_2$ was used to generate OD and measure the rate coefficient ($k_7$) are displayed in Fig. 5. Following the same procedure as outlined above to subtract the contribution of $CH_3C(O)OH$ to the OH decay constant (but using our value of $k_7$ for reaction between OD and $CH_3C(O)OH$) we derive values of $k_5[CH_3C(O)OOH] + k_d$ versus $[CH_3C(O)OOH]$. These are plotted in Fig. 6. From Table S2 we see that, within experimental scatter) the rate coefficient for reaction of OH and OD with $CH_3C(O)OOH$ are identical, and we conclude that OH-reformation via Reactions (R4b + R11) is not responsible for the divergence between our low rate coefficient and the value literature value. Theoretical calculations (section 4.5) also indicate that the reformation of OH in this system is energetically disfavoured.





### 4.4.3 Secondary reactions of OH

As the contribution of $CH_3C(O)OOH$ to the overall loss rate of OH is small, there is potential for overestimation of the rate coefficient if OH can react with products. In this case, we consider the reactions of OH with $CH_3$, which is formed in the photolysis of $CH_3C(O)OOH$ (R10, R15) and in the dominant loss process for OH, reaction with $CH_3C(O)OH$ (R14, R15),

respectively. OH may also react with the $CH_3C(O)O_2$ radical (R17), formed in the title reaction:

| | | | |
|---|---|---|---|
| $CH_3C(O)OOH + h\nu$ | $\rightarrow$ | $CH_3C(O)O + OH$ | (R10) |
| $OH + CH_3C(O)OH$ | $\rightarrow$ | $CH_3C(O)O + H_2O$ | (R14) |
| $CH_3C(O)O$ | $\rightarrow$ | $CH_3 + CO_2$ | (R15) |
| $OH + CH_3 + M$ | $\rightarrow$ | $CH_3OH + M$ | (R16) |
| 350 $OH + CH_3C(O)O_2$ | $\rightarrow$ | $HO_2 + CH_3C(O)O$ | (R17a) |
| | $\rightarrow$ | $CH_3C(O)OOOH$ | (R17b) |

The rate coefficient for reaction of OH with $CH_3$ is rapid, with a room temperature value close to $1 \times 10^{-10}$ $cm^3$ $molecule^{-1}$ $s^{-1}$ (Baulch et al., 2005). There appear not to be any kinetic studies of the reaction between OH and $CH_3C(O)O_2$ but, by analogy to $OH + CH_3O_2$ and $OH + C_2H_5O_2$ (Assaf et al., 2018; IUPAC, 2020) R17 will also have a rate coefficients close to $1 \times 10^{-10}$

$cm^3$ $molecule^{-1}$ $s^{-1}$. In order to assess the role of reactions R16 and R17, we performed numerical simulations of the chemistry subsequent to the generation of OH (and thus $CH_3$) in the photolysis of $CH_3C(O)OOH$ / $CH_3C(O)OH$ mixtures. The simulations were initiated with the concentrations of $CH_3C(O)OOH$, $CH_3C(O)OH$ and OH listed in Table S2. As the decomposition of $CH_3C(O)O$ to $CH_3 + CO_2$ is rapid, we set the initial $CH_3$ concentration equal to that of OH. Along with R16 and R17, we considered inter-radical reactions (e.g. self- and cross-reactions of $CH_3$, $HO_2$ and $CH_3C(O)O_2$) as listed in

Table S3.

For each set of experimental conditions, simulations were carried out in which $k_4$ was varied between zero and $3 \times 10^{-13}$ $cm^3$ $molecule^{-1}$ $s^{-1}$. The simulated decays of OH thus obtained were fitted to an exponential function to obtain the total decay constant, from which the contribution of $CH_3C(O)OH$ was subtracted ($k_6[CH_3C(O)OH]$), as in the experimental data. The results of the simulations are displayed in Fig. 7 along with one set of experimental data obtained at 298 K. Immediately

apparent from the simulations is that values of $k_4 \geq 3 \times 10^{-14}$ $cm^3$ $molecule^{-1}$ $s^{-1}$ over predict the measured slope. Indeed, setting $k_4$ to zero gives the closest agreement between simulation and measurement. A better match between observation and simulation could be obtained by either reducing the initial OH concentration (and thus those of $CH_3$ and $CH_3C(O)O_2$) or lowering the rate coefficients for R16 and R17. The simulated loss of OH was mainly (>90%) via reaction with $CH_3$, which reflects the fact that only a small fraction of OH generated reacts with $CH_3C(O)OOH$ to form $CH_3C(O)O_2$.

Given the uncertainty associated with the determination of the initial radical concentration (based on laser fluence) and with the rate coefficients of the inter-radical reactions involved, it is not possible to use the simulations to correct the experimental data. Instead, recognising that a large fraction of the OH decay constant may be due to unwanted secondary processes, we prefer to quote the value of $k_4$ obtained experimentally as an upper limit.





### 4.4.4 Presence of H$_2$O$_2$ impurity

As indicated in section 4.1.2, the CH$_3$C(O)OOH / CH$_3$C(O)OH mixture is actually an equilibrium mixture containing H$_2$O$_2$ and H$_2$O (R11). Analysis of head-space samples of CH$_3$C(O)OOH and H$_2$O$_2$ using methods, indicate that H$_2$O$_2$ is present at $\approx$ 1% the concentration of CH$_3$C(O)OOH (see section 4.1.2). The IR absorption cross-sections of H$_2$O$_2$ are generally too weak to detect low level impurities so we were unable to unambiguously detect and quantify H$_2$O$_2$ during our kinetic measurements. However, unlike CH$_3$C(O)OOH, H$_2$O$_2$ reacts rapidly with OH, with a rate coefficient of $1.7 \times 10^{-12}$ cm$^3$

molecule$^{-1}$ s$^{-1}$ at 298 K. Initially assuming that $k_4 = 3.2 \times 10^{-14}$ cm$^3$ molecule$^{-1}$ s$^{-1}$ as derived above from the PLP-PLIF experiments would imply that a 1% H$_2$O$_2$ "impurity" in our CH$_3$C(O)OOH sample would result in an overestimation of $k_4$ by ~50%. Together with the considerations of secondary, radical chemistry discussed in section 4.4.2, this leads us to interpret our measurement of $k_4$ as an upper limit and we prefer to quote a value of $k_4 \leq 4 \times 10^{-14}$ cm$^3$ molecule$^{-1}$ s$^{-1}$.

This upper-limit is a factor ~300 lower than the single, previous experimental determination (Wu et al., 2017) who used a

relative-rate technique, which, in principle, offers the advantage that absolute concentrations of CH$_3$C(O)OOH need not be known as long as CH$_3$C(O)OOH and the reference reactant are removed solely via reaction with OH and neither are reformed. However, the relative-rate method does not lend itself readily to the study of this reaction, especially when the 254 nm photolysis of H$_2$O$_2$ is used as OH source, which results both in the photolysis of CH$_3$C(O)OOH and in formation of HO$_2$, which via reactions with CH$_3$C(O)O$_2$ can result in reformation of CH$_3$C(O)OOH. These issues were recognised by Wu et al.

(2017) and corrections applied to take both into account, which resulted in a slight increase in the rate coefficient. In some initial relative-rate experiments in our laboratory, we were unable to derive consistent results as the large affinity of CH$_3$C(O)OOH for surfaces combined with its desorption from the walls during photolysis was too variable to allow analysis of the data.

In our theoretical study (section 4.5), we examine the reaction in detail and show that that the low rate constant we measured

with the PLP-PLIF technique is in good agreement with the predictions.

### 4.5 Theoretical prediction of $k_4$ and the reaction mechanism

The potential energy surface for the CH$_3$C(O)OOH + OH reaction is shown in Fig 8. The addition of OH radicals on a carbonyl double bond is known to have a high barrier and a negligible contribution, and is ignored in this work (Anglada, 2004; De Smedt et al., 2005; Rypkema and Francisco, 2013; Vandenberk et al., 2002). The H-abstraction reactions proceed

through a pre-reaction complex, and feature two protruding barriers for H-abstraction at energies of 2.99 and 3.91 kcal mol$^{-1}$ above the free reactants, corresponding to the abstraction of the per-acetic H-atom and the methyl H-atoms, respectively. The products are formed in a post-reaction complex that quickly dissociates to the free products. The rate coefficients calculated are found to be low, with a value of $k_4 = 3 \times 10^{-14}$ cm$^3$ molecule$^{-1}$ s$^{-1}$ at 298 K. At 298 K, the branching ratio $k_{4a}$ / $k_4$ is predicted to be 0.78 and abstraction of the per-acetic H-atom dominates across the temperature range 200-450 K. Abstraction

of the methyl H-atoms ranges from 10 % at 200 K to 38 % at 450 K. The temperature dependence of the overall rate



coefficient is given by $k_4 = 3.16\times10^{-46}\ T^{10.90}\ \exp(3447\ K/T)$ cm$^3$ molecule$^{-1}$ s$^{-1}$, with $k_{4a}(T) = 1.43\times10^{-43}\ T^{9.87}\ \exp(3287\ K/T)$ cm$^3$ molecule$^{-1}$ s$^{-1}$ and $k_{4b}(T) = 9.65\times10^{-47}\ T^{11.10}\ \exp(3000\ K/T)$ cm$^3$ molecule$^{-1}$ s$^{-1}$. At the level of theory used, the expected uncertainty is a factor of 2 to 3.

The theoretical predictions of $k_4$, $k_{4a}$ and $k_{4b}$ are plotted along with the experimental data from this work and that of Wu et al.
(2017) in Fig. 9. We also indicate the value of $k_4$ (based on comparison with CH$_3$OOH) that is presently used in the Master Chemical Mechanism. Our theoretical work shows that the Arrhenius plot for this reaction is highly curved, with a positive temperature dependence above room temperature, and a negative T-dependence below 280 K. At lower temperatures, abstraction of the per-acetic H-atom is dominant, but at higher temperatures the abstraction of methyl H-atoms through the higher-energy transition state rises in importance and is expected to become dominant at even higher temperatures. Similar
to the reaction between OH and acetic acid (De Smedt et al., 2005; Khamaganov et al., 2006), the curvature in the Arrhenius plot is due to the formation of the pre-reaction complex and subsequent tunneling to the products. With decreasing temperatures, the complex is increasingly populated with a longer lifetime, capturing ever-more (per-)acetic acid + OH complexes and allowing them to tunnel through the barriers at energies below the reactant energies, leading to a negative T-dependence of the rate coefficient. At high temperatures, the lifetime of the pre-reaction complex is too short for effective
tunneling, and the reaction proceeds predominantly over the protruding barriers leading to a traditional positive T-dependence.

With acetic and peracetic acids having similar mechanisms, this does not yet explain why the reaction with per-acetic acid + OH is so much slower than the reaction of acetic acid + OH, despite the fact that the acidic H-abstraction barrier height for CH$_3$C(=O)OH, 3.3 kcal mol$^{-1}$ (De Smedt et al., 2005), is comparable within ~0.3 kcal mol$^{-1}$ to that for CH$_3$C(=O)OOH, 2.99
kcal mol$^{-1}$ (this work). The critical difference lies in the H-bonding in the reactant and pre-reaction complex. In CH$_3$C(O)OH, the H-bond between the acidic H-atom and the carbonyl oxygen is very long, ~2.25 Å, and thus weak. In CH$_3$C(O)OOH acid in contrast, this H-bond is very short, 1.88 Å, and strong. Acetic acid can thus act without a penalty as a bidentate ligand for OH, forming two strong H-bonds leading to a planar complex with 7.3 kcal mol$^{-1}$ stability (De Smedt et al., 2005). CH$_3$C(O)OOH, in contrast, can either form a complex with only a single H-bond on OH, or needs to break the
strong per-acidic bond to form a geometrically unfavourable, non-planar complex with two OH hydrogen bonds. The peracidic complex is thus significantly less stable, by $\approx$ 4.0 kcal mol$^{-1}$, where the energy of the doubly H-bonded OH radical complex is 0.5 kcal mol$^{-1}$ higher than that of the most stable single-H-bonded complex retaining the peracidic H-bond. At room temperature, this weaker bonding decreases the lifetime of the pre-reactive complex by over 2 orders of magnitude compared to acetic acid, significantly reducing its equilibrium concentration and its ability to tunnel slowly through the
barrier, leading to a slower, reduced product formation rate compared to acetic acid, with more of the pre-reaction complexes being formed simply re-dissociating to the free reactants. Concomitantly, the deeper complex well for acetic acid + OH allows this latter reaction to show a negative T-dependent rate coefficient by sustained tunneling even at higher temperatures, up to ~500 K (Khamaganov et al., 2006), well beyond our predictions of a minimum around 270 K for per-acetic acid. Furthermore, we calculate a slightly wider energy barrier for per-acetic acid, with a 1700 cm$^{-1}$ imaginary frequency,





compared to that reported for acetic acid, 2000 cm$^{-1}$ (De Smedt et al., 2005), which further limits tunneling for peracetic acid compared to acetic acid.

The strong H-bond in peracetic acid also make its H-abstraction reactions slower than in alkylhydroperoxides such as CH$_3$OOH. These ROOH compounds can easily form complexes with OH radicals, and the H-abstraction transition states are thus submerged by up to 1 kcal mol$^{-1}$ below the free reactants (Anglada et al., 2017), and even the somewhat less favourable

H$_2$O$_2$ + OH reaction has energy barriers only ~1 kcal mol$^{-1}$ above the reactants (Buszek et al., 2012). This enables the ROOH + OH reactions to proceed substantially faster than CH$_3$C(O)OOH + OH.

Given that the slow abstraction of the peracetic H-atom is a feature of the –C(O)OOH moiety, and that the abstraction reaction is not influenced unduly by other functionalities in the molecules, we propose that the site-specific abstraction rate coefficient can be generalized to all peracids, and used in group-additive structure-activity relationships. Only for long-chain

oxygenated molecules, where an oxygenated group can reach to the –C(O)OOH group and influence the H-bonding with OH, can one expect a non-negligible deviation in the site-specific rate.

## 4.6 Atmospheric Implications

Our experimental and theoretical results indicate that the reaction of CH$_3$C(O)OOH with OH has a rate constant of ~3 × 10$^{-14}$ cm$^3$ molecule$^{-1}$ s$^{-1}$ at temperatures prevalent in the lowermost atmosphere (i.e. in the boundary layer at mid-latitudes),

doubling to ~ 6 × 10$^{-14}$ cm$^3$ molecule$^{-1}$ s$^{-1}$ at temperatures close to 230 K as found in e.g. the upper troposphere. Assuming a global averaged OH abundance of 1 × 10$^6$ molecule cm$^{-3}$, these rate coefficients imply a lifetime of CH$_3$C(O)OOH with respect to degradation by OH of between about 6 months and 1 year. The dominant products of the OH-initiated degradation of CH$_3$C(O)OOH in air are the acetylperoxy radical, CH$_3$C(O)OO, the fate of which is described in section 1, and includes formation of PAN or reformation of CH$_3$C(O)OOH. In air, the minor CH$_2$C(O)OOH product of reaction (R4b) is expected to

add O$_2$, forming a per-acetic acid peroxy radical, OOCH$_2$C(O)OOH, which will also undergo reactions with NO, RO$_2$ and HO$_2$. Throughout much of the atmosphere, its dominant fate will be reaction with NO to form an alkoxy radical, OCH$_2$C(O)OOH which will quickly decompose to CH$_2$O, CO$_2$ and OH (Vereecken and Peeters, 2009).

CH$_2$C(O)OOH + O$_2$ + M          →          OOCH$_2$C(O)OOH                                                        (R14)

OOCH$_2$C(O)OOH + NO          →          OCH$_2$C(O)OOH + NO$_2$                                                        (R15)

OCH$_2$C(O)OOH          →          HCHO + CO$_2$ + OH                                                        (R16)

However, given the low rate coefficient for reaction of CH$_3$C(O)OOH with OH, other loss processes are likely to dominate its atmospheric fate; these are wet and dry deposition, uptake to aerosols and photolysis so that its lifetime will be given by:

$$\tau(\text{CH}_3\text{C(O)OOH}) = \frac{1}{k_4[\text{OH}] + J + k_{dep} + k_{het}}$$                                                        (4)

where $J$ is the first-order rate constant for photolysis by actinic radiation, $k_{dep}$ is the effective loss rate constant for removal

by deposition and $k_{het}$ is the loss rate constant for heterogeneous uptake to aerosol particles. The rate at which CH$_3$C(O)OOH will deposit to surfaces in the boundary layer is given by its deposition velocity and the boundary-layer height. Crowley et al.



(2018) have assessed the terms $k_{dep}$ (for dry deposition) and $k_{het}$ for a summertime, mid-latitude, forested environment. Based on observations of $CH_3C(O)OOH$ and $H_2O_2$, solubilities of $CH_3C(O)OOH$ and $H_2O_2$ (Sander, 1997) actinic flux measurements, the UV-absorption spectrum of $CH_3C(O)OOH$ (Orlando and Tyndall, 2003) aerosol surface areas and an experimental uptake coefficient (Wu et al., 2015) they derived values of $k_{dep} \sim 3\text{-}5 \times 10^{-5}$ s$^{-1}$, $k_{het} \sim 5 \times 10^{-6}$ s$^{-1}$ and $J \sim 5 \times 10^{-7}$ s$^{-1}$ at local noon and concluded that, in the absence of rain, dry-deposition is the dominant loss-process in the boundary layer, followed by reaction with OH, the latter based on an OH rate coefficient of $3\text{-}5 \times 10^{-12}$ cm$^3$ molecule$^{-1}$ s$^{-1}$ as used in the Master Chemical Mechanism.

Above the boundary layer the loss of $CH_3C(O)OOH$ via deposition and heterogeneous uptake to aerosol are less significant so that reaction with OH and photolysis will define its lifetime. A photolysis rate constant ($J$-value) of $\approx 5 \pm 1 \times 10^{-7}$ s$^{-1}$ for $CH_3C(O)OOH$ in the free and upper troposphere results in a lifetime of $\sim$ 3-4 weeks (Orlando and Tyndall, 2003). We note however, that estimates of the photolysis rate constant are based on a single absorption spectrum measured to date (Orlando and Tyndall, 2003) and the assumption of a unity photodissociation quantum yield throughout the UV-absorption spectrum, which remains unconfirmed by experiment or theory.

## 5 Conclusions

Both experimental and theoretical studies of the reaction between OH and $CH_3C(O)OOH$ firmly establish that this is a slow process. The experimental work shows the rate coefficient is $< 4 \times 10^{-14}$ cm$^3$ molecule$^{-1}$ s$^{-1}$ at 298 K, consistent with the theory derived, temperature dependent rate coefficients between 3 and $6 \times 10^{-14}$ cm$^3$ molecule$^{-1}$ s$^{-1}$ for the entire troposphere. The low rate coefficient is rationalised in terms of a weakly-bound (short lived) pre-reaction complex combined with a sufficiently broad reaction barrier to reduce product formation by tunnelling. The site-specific rate coefficient for H-abstraction from the –C(O)OOH moiety can be generalized to most reactions of OH with peracids, which will thus also be slow. The rate coefficient for the OH-reaction is thus at least two orders of magnitude lower than previously reported and implies that the lifetime of $CH_3C(O)OOH$ is dominated by deposition processes (notably dry deposition) in the boundary layer and photolysis in the free and upper troposphere, with OH-initiated degradation playing a minor role. The boundary layer lifetime is expected to be of the order of 1 day, increasing to weeks in the free and upper troposphere. The longer than previously assumed chemical lifetime of $CH_3C(O)OOH$ and probably of other peroxides increase their potential to contribute to secondary organic aerosol formation.

**Acknowledgements**

We thank Dirk Dienhart for measurement of $H_2O_2$ and $CH_3C(O)OOH$ in the headspace of our $CH_3C(O)OOH$ sample. We thank Geoff Tyndall, John Orlando and Tim Wallington for helpful discussions about the IR-spectrum of $CH_3C(O)OOH$ and for providing the NCAR spectrum reported by Orlando et al. in 2000.





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





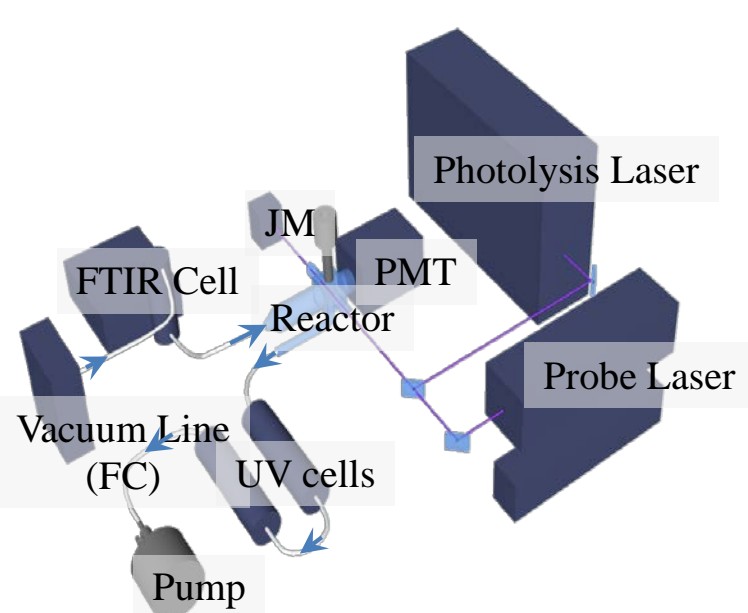

**Figure 1.** PLP-LIF experimental setup. PMT = photomultiplier, JM = Joule meter, FC = flow controller. The IR and UV absorption cells are at room temperature. Photolysis Laser = Excimer Laser (Compex 205 F, 248 nm), Probe Laser = YAG-pumped dye laser (Quantel Brilliant B and Lambda Physik Scanmate, 281.99 and 287.68 nm).




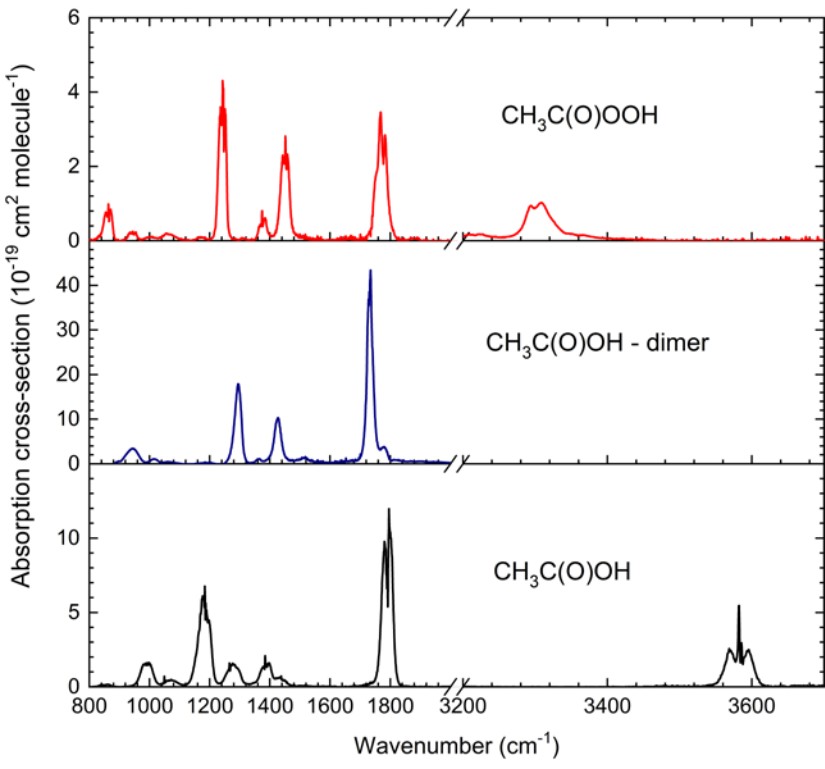

**Figure 2.** IR absorption cross-sections obtained in the long-path absorption cell. A comparison of the CH₃C(O)OOH spectrum with the literature is given in Fig. S2


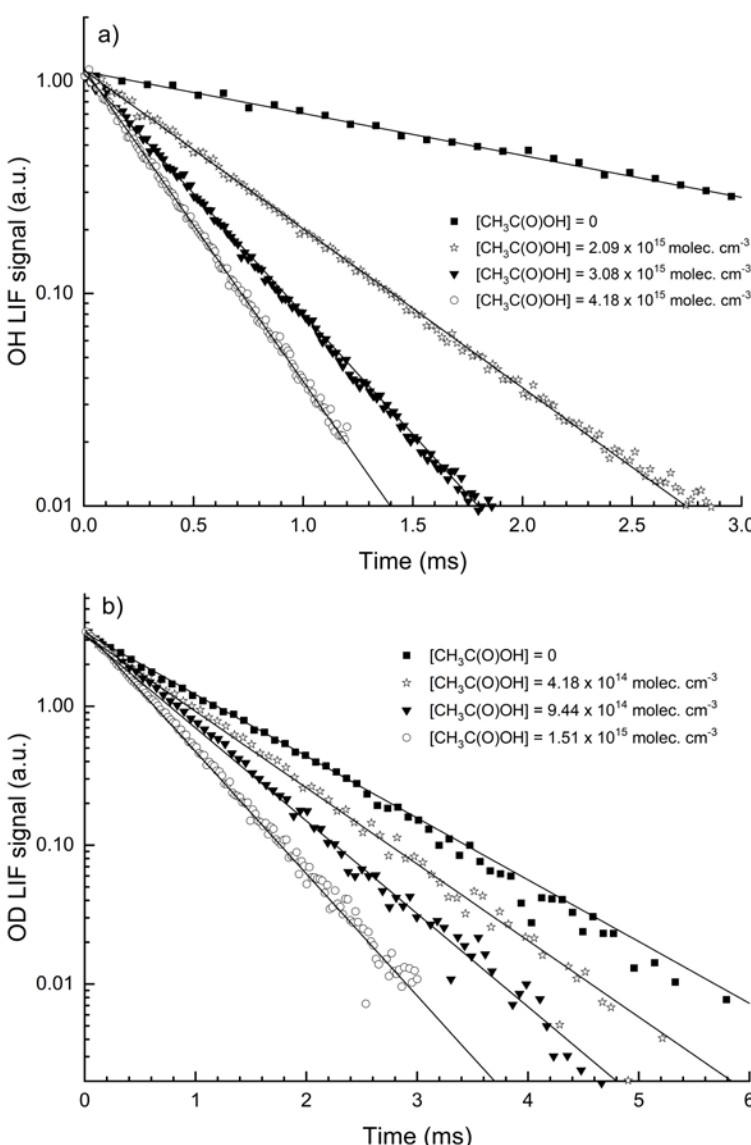

**Figure 3.** Exponential decay of the OH (a) and OD (b) LIF signals in 150 Torr $N_2$, at 293 K, and at four different $CH_3C(O)OH$ concentrations. OH was generated by the photolysis of $H_2O_2$, OD was generated by the photolysis of $DNO_3$ at 248 nm. The lines are fits to the datasets using Equation 1.





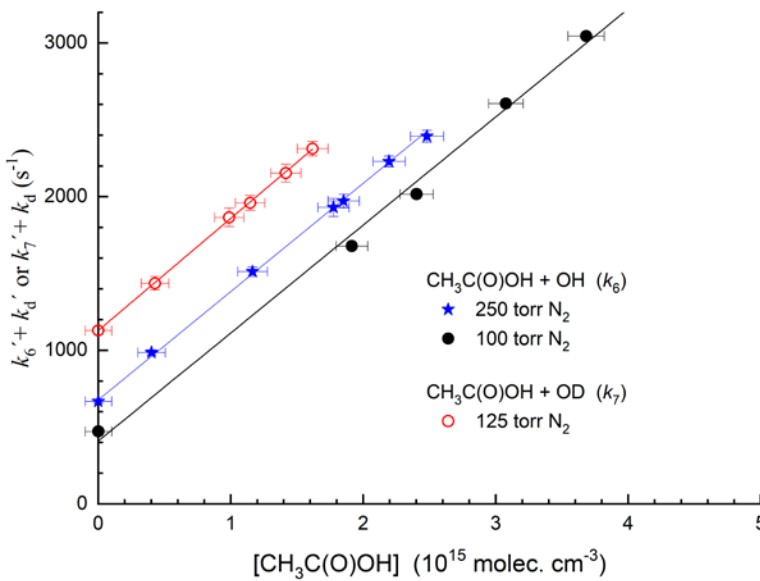


**Figure 4.** Plots of $k'$ vs [CH$_3$C(O)OH] from the decays of OH and OD at different pressures of N$_2$ and 295 K. The lines are least-squares fits to the data using Eq. 2. Error bars are 2σ statistical only. The different intercepts are due to use of different concentrations of H$_2$O$_2$ (OH source) or DONO$_2$ (OD

source).





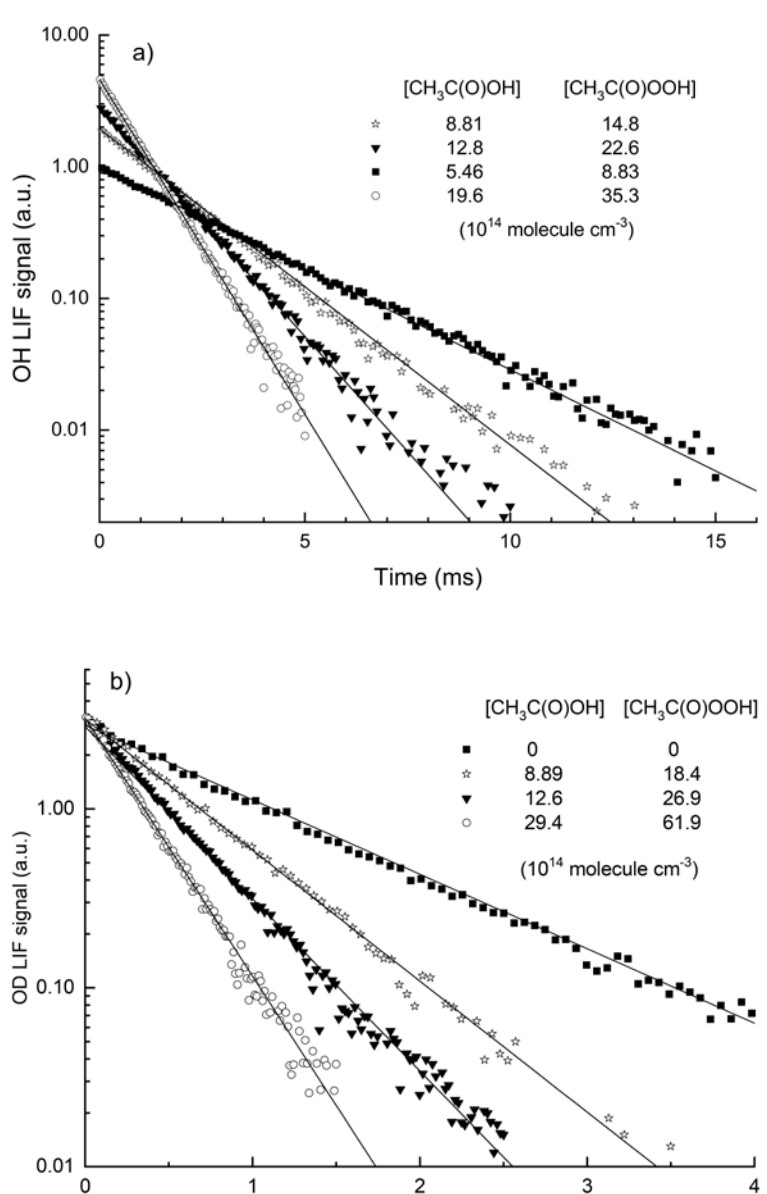


**Figure 5.** a) Exponential decay of the OH LIF-signal in the presence of CH₃C(O)OOH and CH₃C(O)OH in ≈ 150 Torr N₂ at 353 K. OH was generated by the photolysis of CH₃C(O)OOH at 248 nm. b) Exponential decay of the OH LIF-signal in the presence of CH₃C(O)OOH and CH₃C(O)OH in N₂ at 298 K. OD was generated from the 248 nm photolysis of DONO₂.





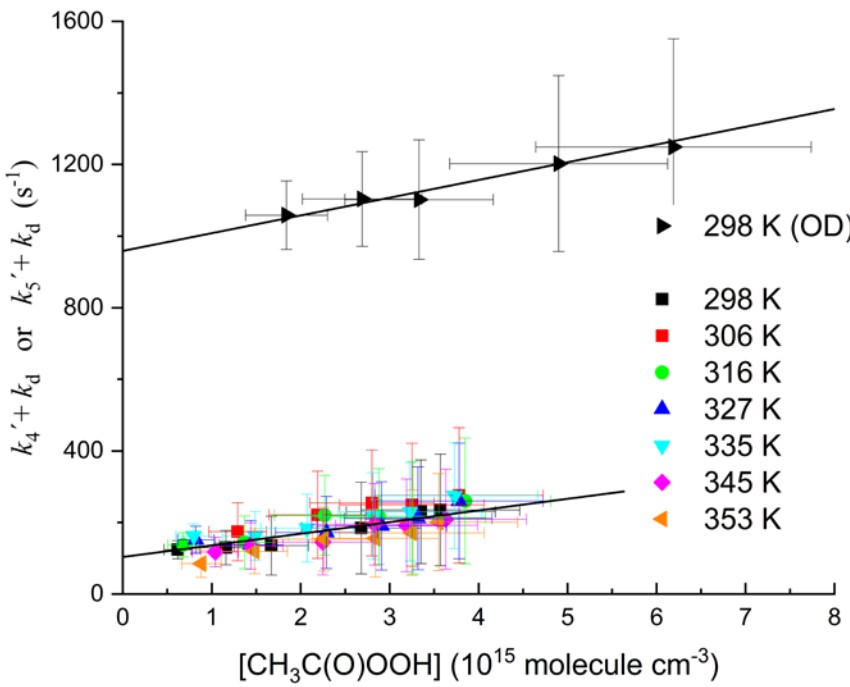

**Figure 6.** Pseudo-first-order rate constant for the loss of OH ($k_4{'}$) or OD ($k_5{'}$) (after subtraction of the
contribution of $CH_3C(O)OH$) versus [$CH_3C(O)OOH$]. The slopes of the solid black lines are $k_4$ (lower
dataset, with intercept $\sim 100$ s$^{-1}$) and $k_6$ (uppermost dataset with intercept $\sim 900$ s$^{-1}$)


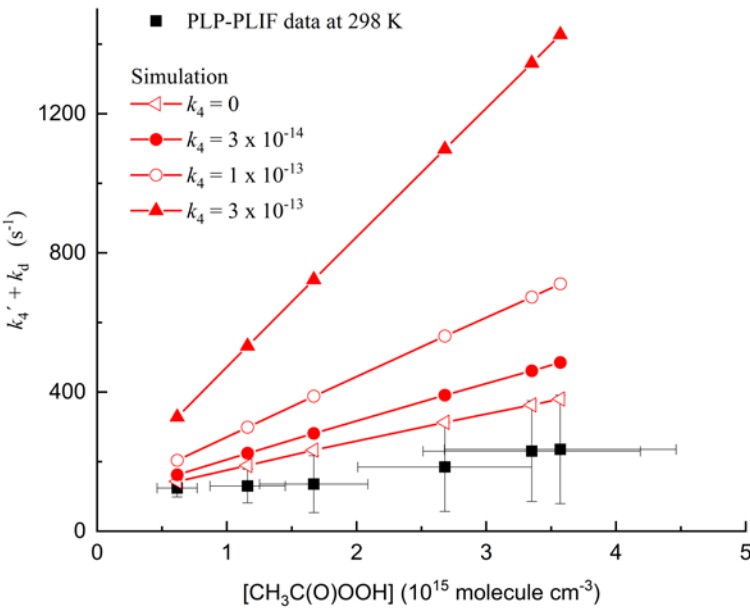

**Figure 7.** Results of 24 simulations (in red) of the chemistry initiated by the photolysis of

$CH_3C(O)OOH$ in the presence of $CH_3C(O)OH$ including reactions of OH with $CH_3$ and $CH_3C(O)O_2$

radicals. As in the experimental data (only those obtained at 298 K are plotted) the contribution of

$CH_3C(O)OH$ to the OH decay constant has been subtracted from each data point. In addition, a

diffusion term of 100 $s^{-1}$ has been added to the simulations so that the same intercept (at zero

$CH_3C(O)OOH$) is obtained.







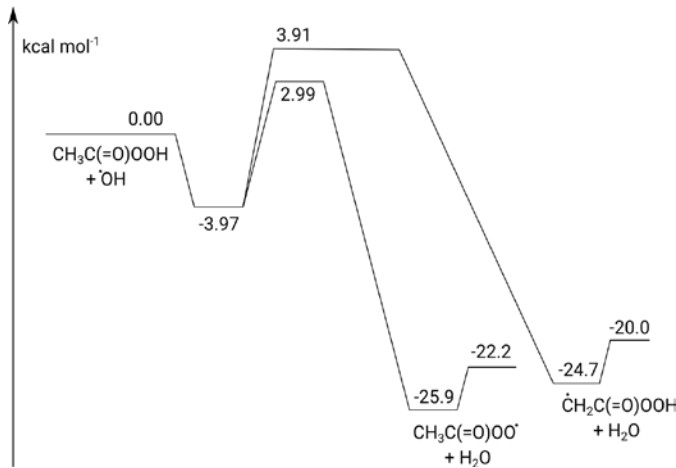

**Figure 8:** ZPE-corrected potential energy surface of the CH₃C(=O)OOH + OH reaction calculated at the CCSD(T)/CBS(DTQ)//M06-2X-D3/aug-cc-pVQZ level of theory.




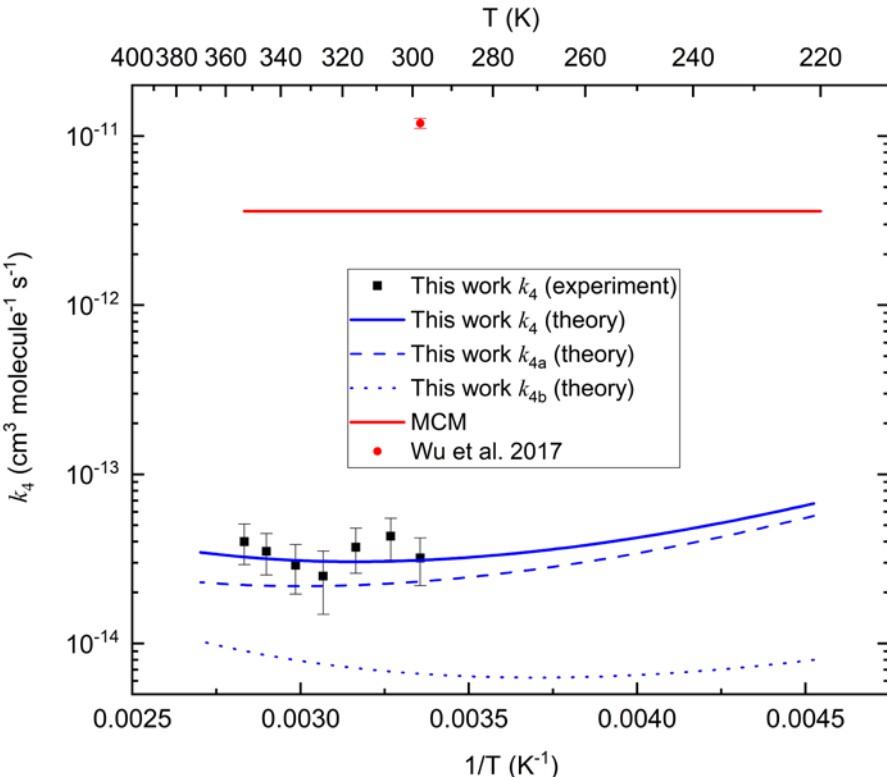

**Figure 9.** Rate coefficients ($k_4$, $k_{4a}$ and $k_{4b}$) for the OH + CH$_3$C(O)OOH reaction. The error bars on the present data-set include uncertainty in the value of $k_4'$ and IR-absorption cross-sections of CH$_3$C(O)OOH. As described in the text, there are several reasons why the experimental data-points should be regarded as upper limits to the rate coefficient.