# Peer review of "Reaction between $\text{CH}_3\text{C(O)OOH}$ (peracetic acid) and OH in the gas-phase: A combined experimental and theoretical study of the kinetics and mechanism"

_Atmospheric Chemistry and Physics, 2020_

## Referee Comment (RC1) · Anonymous Referee #1 · 22 Jul 2020

GENERAL COMMENTS This manuscript revisits the gas-phase kinetics of the OH+CH3C(O)OOH (PAA) reaction as a function of temperature (298-353 K) by an absolute kinetic method. Several additional experiments were performed to check for a potential interference of the OH-reformation (via the study of OD + PAA reaction) and a kinetic numerical simulation was also carried out to quantify the role of secondary OH-reactions. The very low rate coefficient measured experimentally for the OH+CH3C(O)OOH reaction is confirmed theoretically in this work, by using the multi-conformer canonical transition state theory. The disagreement of more than two or-

ders of magnitude with the results from the relative kinetic study by Wu et al. (2017) is discussed. The theoretical temperature-dependence of the rate coefficient of the tittle reaction shows a significant curvature in the Arrhenius plots which is not seen experimentally due to the large uncertainties in k. Despite the calculated negative T-dependence of k, the contribution of the title reaction to the atmospheric removal of CH3C(O)OOH above the boundary layer is not significant with respect to its photolysis in the actinic region, but with a longer lifetime of CH3C(O)OOH (around weeks in the free and upper troposphere) than previously assumed. Finally, the reaction mechanism is investigated theoretically concluding that the importance of CH3C(O)O radical formation. The present comprehensive kinetic study remarks the kinetic complexity of the acetic/peracetic system and the experimental efforts that have to be done to get a reliable rate coefficient. The combination of experimental studies with computation is of great aid to elucidate observed discrepancies or to get deep insights into the reaction mechanisms. Both the experiments and calculations are carefully performed and the paper is generally well structured. In my opinion, the results from this study are of high interest for improving the atmospheric chemical models. Thus, I recommend the publication of this manuscript in the Atmospheric Chemistry and Physics journal after addressing the specific comments/ suggested changes that, in my opinion, need to be included for improving it.

SPECIFIC COMMENTS/SUGGESTIONS 1) Introduction It is said that the branching ratio (BR = k1a / k1) is 0.37 $\pm$ 0.09 at 298 K, while it is 0.31 at 240 K temperatures, stating that the branching ratio to form CH3C(O)OOH decreases with temperature. Well, as no uncertainties are given for the BR at 240 K, I think that it lies within the uncertainties of BR at 298 K.

2) Experimental section The pressure in the reactor was monitored with 10, 100 and 1000 Torr capacitance manometers. Can you provide in the text (and in Tables S1 and S2) the pressure range in the kinetic experiments?

The sentence "The CH3C(O)OH-dimer spectrum was measured in the small cell (path-

length 15 cm) using a similar procedure but using higher pressures of CH3C(O)OH + CH3C(O)OH-dimer (up to 1.36 Torr) to favour dimer formation" is not clear. What is the highest pressure of the mixture acetic acid/dimer? 1.36 Torr in 700 Torr of N2? At the beginning of the section it is stated that "in total typically 3-18 Torr". Please confirm this.

Measurement of H2O, H2O2 and DONO2 concentrations: It is clear that acetic and peracetic acids were quantified by FTIR spectroscopy and that H2O was measured by IR relative to VUV spectroscopy at 185 nm, but what about H2O2 and DONO2? Were they measured by IR spectroscopy too?

3) IR absorption cross sections As the authors known, the absorption cross sections in the IR region are usually expressed in base 10, that is that the absorbance is defined as the log10 of the intensities ratio at a certain wavenumber. In contrast, in the UV region the absorption cross sections are usually expressed in base e, that is that the absorbance is defined as ln of the intensities ratio. Please state in the text.

The authors comment that the IR absorption spectra of H2O in the PAA samples was obtained relative to its VUV-absorption 185 nm. What is the contribution of H2O2 to the total absorption at 185 nm? Is it negligible?

4) Kinetics of OH/OD + CH3C(O)OH reactions The title of section 4.2 is not only "OH + CH3C(O)OH". I suggest this change: "4.2 OH/OD + CH3C(O)OH: Determination of k6 and k7 at 298 K". In some parts of this section refer only a reference to the OH+acetic acid reaction exclusively is made. For example, in equations (1) and (2) or in the sentence "Typically, the values of [CH3C(O)OH] varied by < 3% during the time required to measure the OH-decay". Please complete or revise.

5) Kinetics of OH + CH3C(O)OOH reactions Please, include what ki' is to clarify the relationship with ki. It is far from this section. Can the authors provide a bit more detail on how the concentration of CH3C(O)OOH is corrected by decomposition of the dimer at high temperatures?

Temperature dependence of k4: Why did not the authors measure k4 at temperatures lower than 298 K of interest in the upper troposphere?

Theory: It is stated that the rate coefficient k4 is calculated in the high-pressure limit. The measured k are also in the HPL? What is the total pressure in the reactor?

Secondary OH-reactions: The OH+CH3 reaction is the main loss process in the system. In the text, a reference to Baulch et al. (2005) is made, while in Table S2 the rate coefficient for this reaction is taken from Sangwan et al. (2012). As the OH+CH3 reaction is pressure dependent, the total pressure of the bath gas used has to be stated to derive k given in Table S3 from Sangwan et al.

MINOR SUGGESTED CHANGES When mentioning a range, please use parenthesis. For example, in (6.17 - 38.5) × 1014 molecule cm-3. Use "rate coefficient" or "rate constant" through the manuscript, but not a mixture of both. Unify "Pseudo-first-order" or "pseudo first-order"; "peracetic" or "per-acetic"; "CH3C(O)OOH" or "CH3C(=O)OOH"; "CH2O" or "HCHO".

Line 13: Include units of k in " ∼6 × 10-14 in the cold upper troposphere". Line 18: Replace "dry-deposition" by "dry deposition" Line 19: Replace "peroxy-acids" by "peroxy acids" Line 24-28: This sentence is too long. It can be re-written as: "Despite the acidic peroxide, peracetic acid (CH3C(O)OOH, PAA), is expected to be the 2nd-most abundant organic peroxide (after CH3OOH) in the troposphere, ambient measurements are relatively scarce. Several atmospheric measurements of PAA were reported in the boundary layer (Crowley et al., 2018; Fels and Junkermann, 1994; He et al., 2010; Liang et al., 2013; Phillips et al., 2013; Walker et al., 2006; Zhang et al., 2010) and from aircraft (Crounse et al., 2006; Wang et al., 2019), indicating that it is present throughout the troposphere." Line 61: Replace k2 by k4. Line 74: peroxy radical chemistry Line 96: infrared is more common than infra-red. Line 105: The sentence is confusing, since OH is not generated from DONO2. I suggest to change it: Laser pulses at 248 nm (âĹij20 ns), provided at 10 Hz by an excimer laser (Compex 205

[Figure]

F, Coherent) operated using KrF, were used for generating OH and OD radicals. In particular, H2O2 was used as the photochemical precursor of OH radicals in the study of the OH+CH3C(O)OH reaction, while DONO2 and PAA were used in the study of OD+PAA and OH+PAA reactions, respectively". It is true that this is specified further in next sections. Line 125: 45-cm long Line 211: a quantitative IR spectrum Line 217: 45-cm path-length absorption cell Line 277: Parenthesis are missing in the value of k7. Line 287-288: "...by scaling a reference spectrum of..." Line 333: Change k7 by k5. Line 338: This sentence is not clear"...and we conclude that OH-reformation via Reactions (R4b + R11)..." In reaction 4b, the CH2C(O)OOH radical is formed and R11 is the equilibrium between CH3C(O)OH(l) and CH3C(O)OOH(l). Are these reactions the ones you refer? Line 354: R17 will also have a rate coefficient close Line 376: Delete "using methods" in "Analysis of head-space samples of CH3C(O)OOH and H2O2 using methods..."

Tables and Figures

Tables S1 and S2: Please order the values of k' and k4' increasing the concentration of acetic and peracetic acids for ease of presentation. In Table S2, I would list the values of k'6+kd instead of k'4+kd, since from them k4 are obtained.

Table S3: Replace the = by an arrow in the reaction OH + CH3C(O)O2 = HO2 + CH3

Figure 5a: Decays are better to be normalized as in Figure 5b or Fig 4.

Figure 9, caption: "as upper limits OF the rate coefficient"

Figure S2: The legend of y-axis is not correct. It is not "integrated band strength", it is plotted the "integrated absorbance" with units of cm-1. "Absorbance" is not a physical unit, but a dimensionless parameter.

Figure S3: The legend of y-axis is not correct. "Absorbance" is a dimensionless parameter by definition, so "arbitrary units" has to be deleted.

[Figure]

[Figure]

---

## Referee Comment (RC2) · Anonymous Referee #2 · 27 Jul 2020

General comments

This paper presents a study of the kinetics and mechanism of the reaction of OH with peracetic acid (CH3C(O)OOH), including both an absolute experimental investigation (298 – 353 K) and a theoretical investigation (200 - 450 K). The results show that the reaction is considerably slower than reported previously in a published 298 K relative rate investigation, and as a result an unimportant loss process for CH3C(O)OOH in the atmosphere. Although abstraction of H from the -C(=O)OOH group is calculated to be the dominant reaction pathway, this is found to be orders of magnitude slower than for

simple -OOH groups in species such as CH3OOH.

This is an important piece of work, providing the first direct determination of the title reaction, which will help improve representation of CH3C(O)OOH chemistry (and that of other peroxy-acids) in atmospheric mechanisms. The experimental and theoretical studies are carefully performed, with systematic consideration of possible complications and interferences in the former being carried out and presented. The study is appropriate for publication in ACP, and the authors should consider and address the comments given below in producing an improved version of the manuscript.

Although the core work is well described and justified, this paper would generally have benefitted from more careful proof-reading prior to submission – and this is the origin of most of the comments given below.

Specific comments

1) Line 27: Should "expected to be" be replaced by "observed to be" or simply deleted?

2) Line 27: "2nd-most" should be "second-most".

3) Line 34: Define PAA (or just use CH3C(O)OOH consistently throughout), and delete either "atmospheric" or "in the atmosphere".

4) Line 38: products should be OH + CH3C(O)O + O2 (or "OH + CH3 + CO2 + O2" if subsequent decomposition of CH3C(O)O is included). CH3O2 is not a direct product of the reaction.

5) Line 46: Again, why not write the actual products of the reaction, CH3C(O)O + NO2? Note that you declare CH3C(O)O as a product of reaction (R10), but seem reluctant to do so for reactions (R1c) and (R3).

6) Lines 49-52: This information seems to tally with IUPAC (2020), but shows little similarity to Atkinson et al. (2006). The IUPAC (2020) citation could also be made less vague. For example, could it link to the specific recommendation, rather than the task

group home page?

7) Line 61: The current MCM version is MCM v3.3.1, for which I believe the primary home is now "http://mcm.york.ac.uk/" - although the information is mirrored at "http://mcm.leeds.ac.uk/MCM/" (not "http://mcm.leeds.ac.uk/MC"). However, the described treatment of OH + CH3C(O)OOH remains the same in MCM v3.3.1.

8) Lines 80-83: The degree to which the formation of CH3C(O)OOH from the reaction between HO2 and CH3C(O)O2 represents a loss of oxidation capacity does not depend on whether the CH3C(O)OOH + OH reaction can compete with deposition. Both OH reaction and deposition are radical neutral (i.e. conserve the number of radicals). Reformation of the lost radicals only results from CH3C(O)OOH photolysis, so it is the extent to which the other loss processes compete with photolysis that is important.

9) Line 95: "whereby" would seem to be the wrong adverb here, because the measurement of CH3C(O)OOH and CH3C(O)OH by IR absorption is not achieved as a result of either the laser photolysis production or LIF detection of OH. The information should probably be divided into two sentences after "(LIF)".

10) Line 245: ".....for the reaction between OH and OD with CH3C(O)OH" should probably be ".....for the reactions of OH and OD with CH3C(O)OH."

11) Line 268: I suggest deleting "the values of".

12) Line 273: Again, the IUPAC (2019) citation could link to the specific recommendation for OH + CH3C(O)OH.

13) Section 4.4.1: Either use PAA (defined somewhere) or CH3C(O)OOH. This section oscillates between the two.

14) Line 334: Should k7 be k5?

15) Line 338: the meaning of "OH regeneration via reactions (R4b + R11)" is not clear, these reactions being:

CH3C(O)OOH + OH = CH2C(O)OOH + H2O (R4b)

CH3C(O)OH(l) + H2O2(l) = H2O(l) + CH3C(O)OOH(l) (R11)

16) Line 339: delete first "value".

17) Line 355: According to Table S3, the role of reaction (R17b) was not assessed.

18) Lines 375-383. Given that the possible impact of the impurity H2O2 + OH reaction is assessed, shouldn't that reaction be included in the mechanism in Table S3, and associated simulations, for completeness?

19) Lines 384-395: This information does not seem to fit in a section entitled "Presence of H2O2 impurity".

20) Line 457: Given the main conclusion of the work, the discussion of the chemistry following the OH + CH3C(O)OOH reaction almost seems redundant. The description of the chemistry of the product formed from the minor channel, OOCH2C(O)OOH, also seems selective. Although the chemistry of the NO reaction is important, is its really the dominant fate throughout much of the atmosphere?

21) Lines 476-478: Given the main message of the work, it is not clear why the discussion in an "atmospheric implications" section returns to a conclusion based on using a high rate coefficient for OH + CH3C(O)OOH – even though the preceding comments about other loss processes remain relevant. The atmospheric implications of this work are that the OH + CH3C(O)OOH reaction is unimportant, and that loss is dominated by photolysis and deposition. In my opinion, Section 4.6 could be re-written to state what the atmospheric implications of this work are more clearly and succinctly. In fact, the subsequent conclusions section (section 5) seems to do that very well, and the sections could be merged.

22) Line 496: The reference here to "other peroxides" should probably more correctly state "other peracids", as the conclusions specifically relate to the -C(O)OOH moiety.

In the abstract, this point becomes generalized by the statement "Similar conclusions can be made for other, saturated peroxy-acids", which may be taken to mean that all saturated peroxy-acids can be regarded as having a one-year lifetime with respect to reaction with OH. Presumably the -C(O)OOH moiety deactivates H abstraction from the first carbon in the R group of RC(O)OOH compounds, but abstraction from other sites remains significant – particularly if the (saturated) peroxy-acid contains other activating groups (e.g. -OH). The associated comments could therefore be qualified to this effect.

23) Line 706: Figure 6 caption should state k5 rather than k6. The much larger intercept for k5 presumably results from the reaction of OD with DONO2. This could be stated somewhere.

24) Table S2: PAA (still not defined) in rows; CH3C(O)OOH in columns.

25) Table S3: Footnote "a" also applies to OH + CH3C(O)OH, OH + CH3C(O)O2 and the final channel of HO2 + CH3C(O)O2. Although not crucial for the simulations, it would be nice if O2 and CO2 were declared as products consistently.

---

## Author Comment (AC1) · 16 Sep 2020

The following contains the comments of the referee (black), our replies (blue) indicating changes that will be made to the revised document (red).

**Reviewer #1**

GENERAL COMMENTS
This manuscript revisits the gas-phase kinetics of the OH+CH3C(O)OOH (PAA) reaction as a function of temperature (298-353 K) by an absolute kinetic method. Several additional experiments were performed to check for a potential interference of the OH-reformation (via the study of OD + PAA reaction) and a kinetic numerical simulation was also carried out to quantify the role of secondary OH-reactions. The very low rate coefficient measured experimentally for the OH+CH3C(O)OOH reaction is confirmed theoretically in this work, by using the multiconformer canonical transition state theory. The disagreement of more than two orders of magnitude with the results from the relative kinetic study by Wu et al. (2017) is discussed. The theoretical temperature-dependence of the rate coefficient of the tittle reaction shows a significant curvature in the Arrhenius plots which is not seen experimentally due to the large uncertainties in k. Despite the calculated negative Tdependence of k, the contribution of the title reaction to the atmospheric removal of CH3C(O)OOH above the boundary layer is not significant with respect to its photolysis in the actinic region, but with a longer lifetime of CH3C(O)OOH (around weeks in the free and upper troposphere) than previously assumed. Finally, the reaction mechanism is investigated theoretically concluding that the importance of CH3C(O)O radical formation. The present comprehensive kinetic study remarks the kinetic complexity of the acetic/peracetic system and the experimental efforts that have to be done to get a reliable rate coefficient. The combination of experimental studies with computation is of great aid to elucidate observed discrepancies or to get deep insights into the reaction mechanisms. Both the experiments and calculations are carefully performed and the paper is generally well structured. In my opinion, the results from this study are of high interest for improving the atmospheric chemical models. Thus, I recommend the publication of this manuscript in the Atmospheric Chemistry and Physics journal after addressing the specific comments/ suggested changes that, in my opinion, need to be included for improving it.
We thank the reviewer for the careful review and the positive assessment of our manuscript.

SPECIFIC COMMENTS/SUGGESTIONS
1) Introduction It is said that the branching ratio (BR = k1a / k1) is 0.37 ± 0.09 at 298 K, while it is 0.31 at 240 K temperatures, stating that the branching ratio to form CH3C(O)OOH decreases with temperature. Well, as no uncertainties are given for the BR at 240 K, I think that it lies within the uncertainties of BR at 298 K.
The IUPAC panel recommend a temperature dependence, which is what we quote. We have amended the text to:
Laboratory studies, summarised by IUPAC, indicate that the overall rate coefficient ($k_1$) for reaction R1 (at 298 K) is $(2 \pm 1) \times 10^{-11}$ cm$^3$ molecule$^{-1}$ s$^{-1}$ and that CH$_3$C(O)OOH is formed with a branching ratio ($k_{1a}$ / $k_1$) of 0.37 ± 0.09 at this temperature. At lower temperatures, such as those found in the upper troposphere, the rate coefficient increases ($k_1$ (240 K) = $3.7 \times 10^{-11}$ cm$^3$ molecule$^{-1}$ s$^{-1}$) while the branching ratio to CH$_3$C(O)OOH  decreases:  $k_{1a}$ / $k_1$ (240 K) = 0.31 (Atkinson et al., 2006; IUPAC, 2020).

2) Experimental section The pressure in the reactor was monitored with 10, 100 and 1000 Torr capacitance manometers. Can you provide in the text (and in Tables S1 and S2) the pressure range in the kinetic experiments?

We have added this information to the methods section and the Tables of results in the SI.

The pressure in the reactor, generally between ~50 and 100 Torr $N_2$ was monitored with 100 and 1000 Torr capacitance manometers (1 Torr = 1.333 HPa).

In section 4.2 we write:

$k_6$ was determined at a total pressure ($N_2$) of 57 and 102 Torr, $k_7$ was examined at 66 Torr ($N_2$).

In section 4.3 we write:

The experiments were conducted at a total pressure of ~100 Torr ($N_2$).

In section 4.4.2 we write:

The results of experiments (at ~57 Torr $N_2$) in which…….

The sentence "The CH3C(O)OH-dimer spectrum was measured in the small cell (pathlength 15 cm) using a similar procedure but using higher pressures of CH3C(O)OH + CH3C(O)OH-dimer (up to 1.36 Torr) to favour dimer formation" is not clear. What is the highest pressure of the mixture acetic acid/dimer? 1.36 Torr in 700 Torr of N2? At the beginning of the section it is stated that "in total typically 3-18 Torr". Please confirm this.

The 1.36 Torr was the pressure in the small optical absorption cell. The 3-18 Torr pressures refer to the mixing line. We have modified the text to avoid confusion.

The $CH_3C(O)OH$-dimer spectrum was measured in the small optical absorption cell ($l$ = 15 cm) using up to 1.36 Torr (dosed directly into the cell) of the $CH_3C(O)OH$ / $CH_3C(O)OH$-dimer mixture to favour dimer formation.

Measurement of H2O, H2O2 and DONO2 concentrations: It is clear that acetic and peracetic acids were quantified by FTIR spectroscopy and that H2O was measured by IR relative to VUV spectroscopy at 185 nm, but what about H2O2 and DONO2? Were they measured by IR spectroscopy too?

We made no attempt to measure the concentrations of $H_2O_2$ and $DONO_2$ optically as these parameters are not required in our analysis.

3) IR absorption cross sections As the authors known, the absorption cross sections in the IR region are usually expressed in base 10, that is that the absorbance is defined as the log10 of the intensities ratio at a certain wavenumber. In contrast, in the UV region the absorption cross sections are usually expressed in base e, that is that the absorbance is defined as ln of the intensities ratio. Please state in the text.

We have added text in section 4.1 (Infrared absorption cross-sections).

Note that all IR-cross sections we quote are "base e".

The authors comment that the IR absorption spectra of H2O in the PAA samples was obtained relative to its VUV-absorption 185 nm. What is the contribution of H2O2 to the total absorption at 185 nm? Is it negligible?

The reference IR spectra were obtained using pure $H_2O$ samples and monitoring absorption at 185 nm and in the IR. There is thus no contribution from $H_2O_2$.

4) Kinetics of OH/OD + CH3C(O)OH reactions The title of section 4.2 is not only "OH + CH3C(O)OH". I suggest this change: "4.2 OH/OD + CH3C(O)OH:
Change made:
4.2 OH/OD + CH$_3$C(O)OH: Determination of $k_6$ and $k_7$ at 298 K

Determination of k6 and k7 at 298 K". In some parts of this section refer only a reference to the OH+acetic acid reaction exclusively is made. For example, in equations (1) and (2) or in the sentence "Typically, the values of [CH3C(O)OH] varied by < 3% during the time required to measure the OH-decay". Please complete or revise.
To avoid repeating almost identical equations we now write:
Similar expressions (switch OD for OH and $k_7$ for $k_6$) apply to the OD experiments.
during the time required to measure the OH or OD-decay,

5) Kinetics of OH + CH3C(O)OOH reactions Please, include what ki' is to clarify the relationship with ki. It is far from this section.
We already write: "Where $k_6´$ and $k_4´$ are the pseudo-first order rate constants for loss of OH via reaction (R6) and (R4), respectively."

Can the authors provide a bit more detail on how the concentration of CH3C(O)OOH is corrected by decomposition of the dimer at high temperatures?
There is no known dimer formation for CH$_3$C(O)OOH, so the reviewer probably refers to acetic acid dimer. We now write:
When the reactor is operated at high temperatures some of the CH$_3$C(O)OH-dimer present in the IR-absorption cell is converted to CH$_3$C(O)OH in the reactor and correction was made to account for this using the temperature dependent equilibrium constant.

Temperature dependence of k4: Why did not the authors measure k4 at temperatures lower than 298 K of interest in the upper troposphere?
The decay of OH which we measure is largely due to reaction with CH$_3$C(O)OH and we have no significant contribution from OH reacting with CH$_3$C(O)OOH. As the rate constant for reaction between OH and CH$_3$C(O)OH increases at lower temperatures it made little sense to conduct experiments at temperatures appropriate for e.g. the upper troposphere.

Theory: It is stated that the rate coefficient k4 is calculated in the high-pressure limit. The measured k are also in the HPL? What is the total pressure in the reactor?
We had omitted to mention the pressure and have added text to section 2.1
The pressure in the reactor, generally between ~50 and 100 Torr N$_2$ was monitored with 100 and 1000 Torr capacitance manometers (1 Torr = 1.333 HPa).
We have also added text to section 4.5:
Given the slow product formation rate, the protruding reaction barriers, and the fast formation and decomposition of the complex, $k_4$ is not expected to show a pressure-dependence and should be at the high-pressure limit under the experimental conditions (50-100 Torr N$_2$).

Secondary OH-reactions: The OH+CH3 reaction is the main loss process in the system. In the text, a reference to Baulch et al. (2005) is made, while in Table S2 the rate coefficient for this reaction

is taken from Sangwan et al. (2012). As the OH+CH3 reaction is pressure dependent, the total pressure of the bath gas used has to be stated to derive k given in Table S3 from Sangwan et al. We have removed the citation to Bauch 2005. The termolecular reaction between OH and $CH_3$ is in fact independent of pressure (of He) between ~8 and 680 Torr (Pereira, 1997) and is thus at the high-pressure limit under our experimental conditions. We have chosen to take the rate constant (in He) from the latest work on this reaction for our simulations. We now cite (Table S3) the Pereira et al study (which covers our pressure range) as well as Sangwan et al.

The rate coefficient for reaction of OH with $CH_3$ is at the high-pressure-limit, with a value close to $1 \times 10^{-10}$ $cm^3$ molecule$^{-1}$ s$^{-1}$ (Pereira et al., 1997; Sangwan et al., 2012) under our experimental conditions.

MINOR SUGGESTED CHANGES
When mentioning a range, please use parenthesis. For example, in (6.17 - 38.5) $\times$ 1014 molecule cm-3.
Changes made as suggested

Use "rate coefficient" or "rate constant" through the manuscript, but not a mixture of both.
We now use "rate coefficient" throughout

Unify "Pseudo-first-order" or "pseudo first-order";
We now use pseudo-first-order throughout

"peracetic" or "per-acetic"; "CH3C(O)OOH" or "CH3C(=O)OOH"; "CH2O" or "HCHO".
We now use peracetic, $CH_3C(O)OOH$ and HCHO throughout

Line 13: Include units of k in " ~6 $\times$ 10-14 in the cold upper troposphere".
Change made as suggested

Line 18: Replace "dry-deposition" by "dry deposition"
Change made as suggested

Line 19: Replace "peroxy-acids" by "peroxy acids"
Change made as suggested

Line 24-28: This sentence is too long. It can be re-written as: "Despite the acidic peroxide, peracetic acid (CH3C(O)OOH, PAA), is expected to be the 2nd-most abundant organic peroxide (after CH3OOH) in the troposphere, ambient measurements are relatively scarce. Several atmospheric measurements of PAA were reported in the boundary layer (Crowley et al., 2018; Fels and Junkermann, 1994; He et al., 2010; Liang et al., 2013; Phillips et al., 2013; Walker et al., 2006; Zhang et al., 2010) and from aircraft (Crounse et al., 2006; Wang et al., 2019), indicating that it is present throughout the troposphere."
The sentence has been modified

Line 61: Replace k2 by k4.
Change made as suggested

Line 74: peroxy radical chemistry
Change made as suggested

Line 96: infrared is more common than infra-red.
We now use infrared throughout

Line 105: The sentence is confusing, since OH is not generated from DONO2. I suggest to change it: Laser pulses at 248 nm (â´Lij20 ns), provided at 10 Hz by an excimer laser (Compex 205F, Coherent) operated using KrF, were used for generating OH and OD radicals. In particular, H2O2 was used as the photochemical precursor of OH radicals in the study of the OH+CH3C(O)OH reaction, while DONO2 and PAA were used in the study of OD+PAA and OH+PAA reactions, respectively". It is true that this is specified further in next sections.
Text changed, we now write:
Pulses of 248 nm laser light ($\sim$20 ns) for OH generation from $H_2O_2$ and $CH_3C(O)OOH$ or OD generation from $DONO_2$ were provided…..

Line 125: 45-cm long
Text changed, we now write:
…by flowing the sample through an absorption cell ($l = 45$ cm) made of glass,

Line 211: a quantitative IR spectrum
Change made as suggested

Line 217: 45-cm path-length absorption cell
Change made as suggested

Line 277: Parenthesis are missing in the value of k7.
Change made as suggested

Line 287-288: ". . .by scaling a reference spectrum of. . ."
Change made as suggested

Line 333: Change k7 by k5.
Change made as suggested

Line 338: This sentence is not clear". . .and we conclude that OH-reformation via Reactions (R4b + R11). . ." In reaction 4b, the CH2C(O)OOH radical is formed and R11 is the equilibrium between CH3C(O)OH(l) and CH3C(O)OOH(l). Are these reactions the ones you refer?
The reactions were wrongly numbered. We now simply state
…and we conclude that OH-reformation is not responsible for the divergence….

Line 354: R17 will also have a rate coefficient close
Change made as suggested

Line 376: Delete "using methods" in "Analysis of head-space samples of CH3C(O)OOH and H2O2 using methods. . ."

Change made as suggested

Tables and Figures

Tables S1 and S2: Please order the values of k' and k4' increasing the concentration of acetic and peracetic acids for ease of presentation.
Change made as suggested

In Table S2, I would list the values of k'6+kd instead of k'4+kd, since from them k4 are obtained.
$k_4$ is obtained from plots of $k_4´+ k_d$ versus [$CH_3C(O)OOH$], this is why we list this parameter.
Table S3: Replace the = by an arrow in the reaction OH + CH3C(O)O2 = HO2 + CH3
Change made as suggested

Figure 5a: Decays are better to be normalized as in Figure 5b or Fig 4.
In Figure 5b, the photolysis of $DONO_2$ is used as OH source and thus the initial amount of OD does not change. In Figure 5a, the amount of OH is variable (as its precursor was $CH_3C(O)OOH$.

Figure 9, caption: "as upper limits OF the rate coefficient"
Text modified

Figure S2: The legend of y-axis is not correct. It is not "integrated band strength", it is plotted the "integrated absorbance" with units of cm-1. "Absorbance" is not a physical unit, but a dimensionless parameter.
This comment refers to S1.
We have corrected the y-axis labelling

Figure S3: The legend of y-axis is not correct. "Absorbance" is a dimensionless parameter by definition, so "arbitrary units" has to be deleted.
Change made as suggested

---

## Author Comment (AC2) · 16 Sep 2020

**Reviewer #2**

General comments. This paper presents a study of the kinetics and mechanism of the reaction of OH with peracetic acid ($CH_3C(O)OOH$), including both an absolute experimental investigation (298 – 353 K) and a theoretical investigation (200 - 450 K). The results show that the reaction is considerably slower than reported previously in a published 298 K relative rate investigation, and as a result an unimportant loss process for $CH_3C(O)OOH$ in the atmosphere. Although abstraction of H from the -C(=O)OOH group is calculated to be the dominant reaction pathway, this is found to be orders of magnitude slower than for simple -OOH groups in species such as $CH_3OOH$. This is an important piece of work, providing the first direct determination of the title reaction, which will help improve representation of $CH_3C(O)OOH$ chemistry (and that of other peroxy-acids) in atmospheric mechanisms. The experimental and theoretical studies are carefully performed, with systematic consideration of possible complications and interferences in the former being carried out and presented. The study is appropriate for publication in ACP, and the authors should consider and address the comments given below in producing an improved version of the manuscript. Although the core work is well described and justified, this paper would generally have benefitted from more careful proof-reading prior to submission – and this is the origin of most of the comments given below.

We thank the reviewer for the careful review and the positive assessment of our manuscript.

Specific comments
1) Line 27: Should "expected to be" be replaced by "observed to be" or simply deleted?
We now write "observed to be"

2) Line 27: "2nd-most" should be "second-most".
Change made as suggested

3) Line 34: Define PAA (or just use $CH_3C(O)OOH$ consistently throughout), and delete either "atmospheric" or "in the atmosphere".
Change made as suggested

4) Line 38: products should be $OH + CH_3C(O)O + O_2$ (or "$OH + CH_3 + CO_2 + O_2$" if subsequent decomposition of $CH_3C(O)O$ is included). $CH_3O_2$ is not a direct product of the reaction.
Change made as suggested

5) Line 46: Again, why not write the actual products of the reaction, $CH_3C(O)O + NO_2$? Note that you declare $CH_3C(O)O$ as a product of reaction (R10), but seem reluctant to do so for reactions (R1c) and (R3). 6)
Change made as suggested

Lines 49-52: This information seems to tally with IUPAC (2020), but shows little similarity to Atkinson et al. (2006). The IUPAC (2020) citation could also be made less vague. For example, could it link to the specific recommendation, rather than the task group home page?
We now cite only the 2020 (online) IUPAC recommendation.

7) Line 61: The current MCM version is MCM v3.3.1, for which I believe the primary home is now "http://mcm.york.ac.uk/" - although the information is mirrored at http://mcm.leeds.ac.uk/MCM/" (not "http://mcm.leeds.ac.uk/MC"). However, the described treatment of OH + CH3C(O)OOH remains the same in MCM v3.3.1.
We now cite only the current MCM version.

8) Lines 80-83: The degree to which the formation of CH3C(O)OOH from the reaction between HO2 and CH3C(O)O2 represents a loss of oxidation capacity does not depend on whether the CH3C(O)OOH + OH reaction can compete with deposition. Both OH reaction and deposition are radical neutral (i.e. conserve the number of radicals). Reformation of the lost radicals only results from CH3C(O)OOH photolysis, so it is the extent to which the other loss processes compete with photolysis that is important.
We now write:
The degree to which the formation of $CH_3C(O)OOH$ from the reaction between $HO_2$ and $CH_3C(O)O_2$ represents a permanent sink of peroxy radicals (and thus loss of oxidation capacity) depends on whether the photochemical degradation of $CH_3C(O)OOH$ to reform organic radicals can compete with deposition processes.

9) Line 95: "whereby" would seem to be the wrong adverb here, because the measurement of CH3C(O)OOH and CH3C(O)OH by IR absorption is not achieved as a result of either the laser photolysis production or LIF detection of OH. The information should probably be divided into two sentences after "(LIF)".
Change made as suggested

10) Line 245: ".....for the reaction between OH and OD with CH3C(O)OH" should probably be ".....for the reactions of OH and OD with CH3C(O)OH."
We now write:
We therefore carried out a set of experiments to measure the rate coefficients for the reactions of OH and OD with $CH_3C(O)OH$.

11) Line 268: I suggest deleting "the values of".
Change made as suggested

12) Line 273: Again, the IUPAC (2019) citation could link to the specific recommendation for OH + CH3C(O)OH.
We now reference the current (2020) version.

13) Section 4.4.1: Either use PAA (defined somewhere) or CH3C(O)OOH. This section oscillates between the two.
All reference to PAA has been removed.

14) Line 334: Should k7 be k5?
Yes, correction made.

15) Line 338: the meaning of "OH regeneration via reactions (R4b + R11)" is not clear, these reactions being:

CH3C(O)OOH + OH = CH2C(O)OOH + H2O (R4b)
CH3C(O)OH(l) + H2O2(l) = H2O(l) + CH3C(O)OOH(l) (R11)
Reference to R4b and R11 has been removed

16) Line 339: delete first "value".
Change made as suggested

17) Line 355: According to Table S3, the role of reaction (R17b) was not assessed.
We now mention only reaction R17a

18) Lines 375-383. Given that the possible impact of the impurity H2O2 + OH reaction is assessed, shouldn't that reaction be included in the mechanism in Table S3, and associated simulations, for completeness?
As we have no in-situ measurement of $[H_2O_2]$ measurement and its contribution (if present at the 1% level) can easily be assessed without numerical simulation we did not include it in the reaction scheme.

19) Lines 384-395: This information does not seem to fit in a section entitled "Presence of H2O2 impurity".
True. We have generated a new section that deals with the comparison.
4.5 Comparison with the previous determination of $k_4$.

20) Line 457: Given the main conclusion of the work, the discussion of the chemistry following the OH + CH3C(O)OOH reaction almost seems redundant. The description of the chemistry of the product formed from the minor channel, OOCH2C(O)OOH, also seems selective. Although the chemistry of the NO reaction is important, is its really the dominant fate throughout much of the atmosphere?
Although (as we conclude a few sentences later) OH will not be an important sink of $CH_3C(O)OOH$, we prefer to keep this text so as to provide a more complete picture of the role of $CH_3C(O)OOH$ in the atmosphere. We have amended the text regarding the dominant fate of $CH_3C(O)OOH$ and have added reactions with $HO_2$ and $RO_2$.
In many regions of the atmosphere (e.g. those impacted by anthropogenic emissions) its dominant fate will be reaction with NO….
In air, the minor $CH_2C(O)OOH$ product of reaction (R4b) is expected to add $O_2$, forming a peracetic acid peroxy radical, $OOCH_2C(O)OOH$, which will also undergo reactions with NO, $RO_2$ and $HO_2$.

| | | | |
|---|---|---|---|
| $CH_2C(O)OOH + O_2 + M$ | $\rightarrow$ | $OOCH_2C(O)OOH$ | (R14) |
| $OOCH_2C(O)OOH + NO$ | $\rightarrow$ | $OCH_2C(O)OOH + NO_2$ | (R15) |
| $OOCH_2C(O)OOH + HO_2$ | $\rightarrow$ | $HOOCH_2C(O)OOH + O_2$ | (R16) |
| $OOCH_2C(O)OOH + RO_2$ | $\rightarrow$ | $OCHC(O)OOH + ROH + O_2$ | (R17a) |
| | $\rightarrow$ | $HOCH_2C(O)OOH + R=O + O_2$ | (R17b) |
| | $\rightarrow$ | $OCHC(O)OOH + RO + O_2$ | (R17c) |

$OCH_2C(O)OOH$ which will quickly decompose to HCHO, $CO_2$ and OH (Vereecken and Peeters, 2009).

| | | | |
|---|---|---|---|
| $OCH_2C(O)OOH$ | $\rightarrow$ | $HCHO + CO_2 + OH$ | (R18) |

21) Lines 476-478: Given the main message of the work, it is not clear why the discussion in an "atmospheric implications" section returns to a conclusion based on using a high rate coefficient for OH + CH3C(O)OOH – even though the preceding comments about other loss processes remain relevant. The atmospheric implications of this work are that the OH + CH3C(O)OOH reaction is unimportant, and that loss is dominated by photolysis and deposition. In my opinion, Section 4.6 could be re-written to state what the atmospheric implications of this work are more clearly and succinctly. In fact, the subsequent conclusions section (section 5) seems to do that very well, and the sections could be merged.

We have removed reference to the high rate coefficient and taken the description of the products formed from OH + $CH_3C(O)OOH$ out of the "atmospheric implications" section and moved them to the "theoretical studies" section.

22) Line 496: The reference here to "other peroxides" should probably more correctly state "other peracids", as the conclusions specifically relate to the -C(O)OOH moiety. In the abstract, this point becomes generalized by the statement "Similar conclusions can be made for other, saturated peroxy-acids", which may be taken to mean that all saturated peroxy-acids can be regarded as having a one-year lifetime with respect to reaction with OH. Presumably the -C(O)OOH moiety deactivates H abstraction from the first carbon in the R group of RC(O)OOH compounds, but abstraction from other sites remains significant – particularly if the (saturated) peroxy-acid contains other activating groups (e.g. -OH). The associated comments could therefore be qualified to this effect.

We agree. The statement was too general as has been removed from the abstract.

23) Line 706: Figure 6 caption should state k5 rather than k6. The much larger intercept for k5 presumably results from the reaction of OD with DONO2. This could be stated somewhere.

$k_6$ replaced with $k_5$

We also added the text:

The larger intercept for the OD reaction is due to reaction with $DONO_2$.

24) Table S2: PAA (still not defined) in rows; CH3C(O)OOH in columns.

PAA is now defined as $CH_3C(O)OH$ in Table S2.

25) Table S3: Footnote "a" also applies to OH + CH3C(O)OH, OH + CH3C(O)O2 and the final channel of HO2 + CH3C(O)O2. Although not crucial for the simulations, it would be nice if O2 and CO2 were declared as products consistently.

Footnote "a" has been added to the appropriate reactions. $CO_2$ and $O_2$ have been added where appropriate.